# Learning What Matters Now: Dynamic Preference Inference under Contextual Shifts

**Xianwei Cao**[1,2]    **Dou Quan**[1]    **Zhenliang Zhang**[2]*    **Shuang Wang**[1]*
[1]School of Artificial Intelligence, Xidian University, Xi'an, 710071, China
[2]State Key Laboratory of General Artificial Intelligence, BIGAI, Beijing, China

zlzhang@bigai.ai, shwang@mail.xidian.edu.cn

## Abstract

Humans often juggle multiple, sometimes conflicting objectives and shift their priorities as circumstances change, rather than following a fixed objective function. In contrast, most computational decision-making and multi-objective RL methods assume static preference weights or a known scalar reward. In this work, we study sequential decision-making problem when these preference weights are unobserved latent variables that drift with context. Specifically, we propose Dynamic Preference Inference (DPI), a cognitively inspired framework in which an agent maintains a probabilistic belief over preference weights, updates this belief from recent interaction, and conditions its policy on inferred preferences. We instantiate DPI as a variational preference inference module trained jointly with a preference-conditioned actor–critic, using vector-valued returns as evidence about latent trade-offs. In queueing, gridworld maze, and multi-objective continuous-control environments with event-driven changes in objectives, DPI adapts its inferred preferences to new regimes and achieves higher post-shift performance than fixed-weight and heuristic envelope baselines[1].

## 1 Introduction

Human behavior is widely modeled as goal-directed and value-driven, but people typically juggle multiple goals and adjust their priorities as circumstances change, rather than acting on a single fixed priority ordering (Simon, 1955; Payne et al., 1993; Carver & Scheier, 2001; Wrosch et al., 2003). We reweight priorities, abandon infeasible objectives, and reorient toward what remains achievable. For instance, as illustrated in Fig. 1, a person waiting in line may initially value fairness and patience, but as hunger escalates and time runs out, they may rationalize cutting ahead. Work on self-regulation, multiple-goal pursuit, and constructed preferences formalizes such behavior as feedback-based goal control and context-dependent weighting of attributes (Payne et al., 1993; Slovic, 1995; Lichtenstein & Slovic, 2006). Yet, despite its importance, computational modeling of *dynamic value preference adaptation* remains underexplored in artificial intelligence and multi-objective decision-making (Roijers et al., 2013; Yang et al., 2019; Agarwal et al., 2022; Basaklar et al., 2023; Liu et al., 2025).

A large literature in cognitive psychology and cognitive science has examined how people regulate goals and reconcile competing motives. Theories of self-regulation and multiple-goal pursuit emphasize feedback-based adjustment of goal importance and effort allocation (Carver & Scheier, 2001; 2004; Wrosch et al., 2003), while multi-attribute and context-dependent choice models show that attribute weights and even preferences themselves can shift with task demands and elicitation formats (Payne et al., 1993; Slovic, 1995; Lichtenstein & Slovic, 2006). Abstracting from these literatures, we conceptually separate two tightly coupled processes: (1) *value appraisal*—forming an internal judgment about what matters most *right now* in a given situation; and (2) *action selection*—choosing behavior conditional on the current value preference. In this paper, we use these

---

*Corresponding authors.
[1]Code is available at: https://github.com/XianweiC/DPI

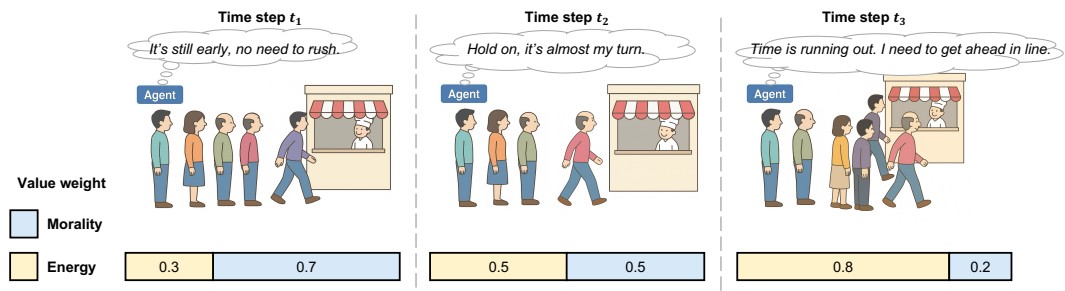

Figure 1: **Adaptive value preference adjustment in a queueing scenario.** At early stages ($t_1$), the agent prioritizes morality and chooses to wait. As the deadline approaches ($t_2$), preferences between morality (M) and energy (E) become balanced. When time is nearly exhausted ($t_3$), energy becomes dominant and the agent rationalizes cutting in line, illustrating dynamic reweighting of values under changing pressures.

terms purely as labels for components of our computational framework: the second has been extensively studied as policy optimization under a given reward or utility function, whereas the first—how an agent constructs and *updates* its value function from experience and context—remains comparatively underexplored in computational models. From a modeling perspective, many psychological and choice-theoretic accounts provide rich *descriptions* of how humans adjust goals and preferences, but they are not directly formulated as algorithms that can be deployed in high-dimensional, partially observable, and non-stationary control problems. In contrast, artificial agents in safety-critical domains must make a stream of sequential decisions from raw observations, under changing resource, time, or risk constraints. This motivates the computational question at the core of our work: **given only vector-valued rewards and partial observations in a non-stationary environment, how can an agent infer and adapt its current trade-off over objectives online in a way that is both effective and interpretable?** We address this question in the language of multi-objective reinforcement learning, using dynamic preference inference as a bridge between cognitive theories of goal regulation and practical control algorithms.

Many computational models of sequential decision-making—including Markov decision processes and modern reinforcement learning agents—assume a fixed and externally specified utility or reward function (Bellman & Kalaba, 1957; Russell & Norvig, 1995). While this assumption simplifies learning and planning, it fails to capture the fluid, context-dependent nature of value trade-offs observed in realistic settings. In multi-objective environments (e.g., efficiency vs. safety, energy vs. morality), agents inevitably face *shifting constraints*: some goals become infeasible, temporarily irrelevant, or disproportionately important as resources, time, or external conditions change (Roijers et al., 2013; Vamplew et al., 2011). Without the capacity for dynamic value reconfiguration, such agents risk either pursuing unattainable goals or neglecting emergent priorities—leading to suboptimal or even catastrophic outcomes in domains such as autonomous driving or healthcare (Amodei et al., 2016; Dulac-Arnold et al., 2021).

To mitigate this limitation, recent work in multi-objective decision-making has developed preference-conditioned policies, where an agent is trained to optimize under different trade-offs specified by a preference vector (Van Moffaert & Nowé, 2014; Yang et al., 2019; Basaklar et al., 2023; Liu et al., 2025). Such approaches enable generalization across static preferences and complement earlier scalarization and envelope methods (Vamplew et al., 2011; Mossalam et al., 2016). However, they typically assume that the preference vector is given. In realistic scenarios, preferences are rarely observed directly: the agent must *infer* its priorities from incomplete, noisy, and evolving perceptual data. This calls for an online preference inference mechanism that is sensitive to environmental cues yet robust to transient noise—a capability that is both cognitively plausible and computationally underexplored.

In this work, we propose a **Dynamic Preference Inference (DPI)** framework that makes this gap explicit. The key idea is to treat the preference vector encoding the relative importance of multiple objectives as a *latent* state that must be inferred online rather than fixed a priori. The agent maintains a posterior distribution $q_\phi(\boldsymbol{\omega}_t \mid s_{t-H+1:t})$ over current preference vector $\boldsymbol{\omega}_t$, parameterized by a recurrent encoder over recent history. Sampling from this posterior captures epistemic uncertainty and lets the agent explore alternative value configurations before acting. A preference-conditioned actor–critic then conditions its policy on $\boldsymbol{\omega}_t$ to adapt behavior as task demands shift. To keep updates stable and interpretable, we regularize $q_\phi$ with a Gaussian prior and a directional alignment term

that encourages preference changes consistent with observed return vectors. DPI thus provides a compact variational Bayesian formulation of value appraisal, in which the agent maintains a belief over "what matters now" and revises it only when recent outcomes provide sufficient evidence.

The main contributions of this work are:

- We formalize a setting where preference weights in environments are *unknown and dynamically varying*, and highlight the challenge of enabling agents to infer and adapt their priorities online from experience.
- We introduce the **Dynamic Preference Inference (DPI)** framework, a computational architecture that jointly learns (i)a probabilistic preference inference model from perceptual history and (ii)a preference-conditioned policy, regularized for stability and interpretability.
- We empirically evaluate DPI in a dynamic Queue environment and a dynamic Maze environment, showing consistent gains over fixed-preference and static-inference baselines in adaptability, robustness, and cumulative performance, and illustrating more interpretable, event-aligned adaptive strategies.

## 2 Related Work

### 2.1 Multi-Objective Decision-Making in Cognitive Science

Human decision-making rarely involves optimizing a single objective in isolation. Instead, individuals continuously negotiate multiple, sometimes conflicting goals such as efficiency, fairness, energy preservation, and social norms. Classical theories of bounded rationality (Simon, 1955) argue that humans rely on satisficing heuristics rather than globally optimal strategies. Dynamic models such as Decision Field Theory (Busemeyer & Townsend, 1993) and Prospect Theory (Kahneman & Tversky, 2013) emphasize that preferences evolve with time pressure, risk, and context. Research on multi-attribute decision-making (Payne et al., 1993; Zanakis et al., 1998) shows that humans flexibly reweight attributes depending on contextual demands—for example, prioritizing efficiency under time pressure or fairness in cooperative settings. Theories of self-regulation and control further highlight that goal pursuit is adaptive, context-sensitive, and autonomy-driven (Shenhav et al., 2013; Deci & Ryan, 1985; 2012). These insights motivate computational frameworks that treat preferences not as fixed constants, but as latent variables dynamically inferred from context.

### 2.2 Computational Approaches to Multi-Objective Decision-Making

Multi-objective reinforcement learning (MORL) provides a principled framework for sequential decision-making with vector-valued rewards. Classical approaches rely on scalarization (Vamplew et al., 2011; Roijers et al., 2013; Agarwal et al., 2022), collapsing the reward vector into a scalar using a fixed preference vector. While effective for static settings, these approaches fail under shifting priorities. Pareto-based methods, such as Envelope Q-learning and Pareto-conditioned policy optimization (Van Moffaert & Nowé, 2014; Yang et al., 2019; Basaklar et al., 2023; Liu et al., 2025), approximate the Pareto front of optimal policies, enabling post-hoc preference selection. However, they still assume externally specified preferences and lack online adaptation. Recent work explores nonlinear and dynamic scalarization (Mossalam et al., 2016; Abels et al., 2019), meta-learning for preference adaptation, and learning from human feedback (Christiano et al., 2017; Ibarz et al., 2018; Ramachandran & Amir, 2007; Fu et al., 2018). Nevertheless, few methods explicitly treat preference weights as latent states to be inferred online. Our approach addresses this gap by framing preference adaptation as a variational inference problem, enabling agents to maintain a belief over preferences and dynamically reweight objectives in response to environmental changes, aligning with psychological theories of adaptive goal regulation.

## 3 Methodology

We ground our study on the assumption of a boundedly rational agent, capable of reappraising what matters in response to situational changes (Simon, 1955; Carver & Scheier, 2001; Kruglanski et al., 2018; Zhang et al., 2024). Human decision-making in dynamic, multi-objective settings rarely follows a single, fixed plan. Instead, we continuously reappraise our goals in light of the current

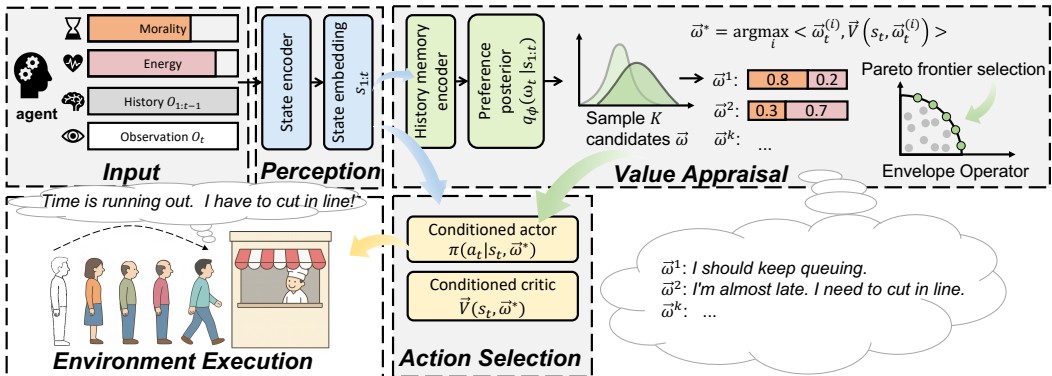

Figure 2: Two-stage cognitive-inspired decision framework. History states are transformed into latent preferences via **Value Appraisal**, which in turn guide the **Action Selection**. The resulting policy drives environment execution, forming a dynamic decision pipeline analogous to human appraisal–action coupling.

situation—deciding not just how to act, but also what matters most right now. This dual process is reflected in cognitive models such as appraisal theory in emotion research (Lazarus, 1991; Scherer, 1999; Frijda, 1993) and dual-process frameworks in psychology (Kahneman, 2011; Stanovich et al., 2000; Kahneman, 2012), where value assessment and action selection are distinct yet tightly coupled.

In this work, we operationalize these principles through a two-module computational agent (Fig. 2):

- **Value Appraisal Module (history → preferences):** From a short history-memory window of states, a recurrent encoder infers a distribution over the latent preference vector $\boldsymbol{\omega}_t \in \Delta^{d-1}$ (a point on the simplex), from which a sample $\hat{\boldsymbol{\omega}}_t$ is drawn for control.

- **Action Selection Module (state, preferences → action):** An actor–critic conditions jointly on the current state and inferred preferences to produce vector value estimates and a policy. At decision time, an on-policy envelope operator evaluates $K$ preference samples and selects the $\hat{\boldsymbol{\omega}}_t$ that maximizes the predicted scalarized value.

- **Stability & Alignment:** Training is regularized by a variational evidence lower bound (ELBO) with a simple prior, a directional alignment between inferred preferences and vector returns, and a self-consistency term anchoring posterior predictions to the envelope-selected $\hat{\boldsymbol{\omega}}_t$. Together, these stabilize preference inference and improve interpretability.

This division is consistent with cognitive models of dual-process decision-making: a working memory–based appraisal system that updates value priorities, and a controller that acts accordingly.

## 3.1 VALUE APPRAISAL AS VARIATIONAL INFERENCE OVER DYNAMIC PREFERENCES

**Cognitive & statistical generative assumption.** Let $\boldsymbol{\omega}_t^* \in \Delta^{d-1}$ denote the (unobserved) preference weights over $d$ objectives at time $t$. We assume a non-stationary latent dynamics $\boldsymbol{\omega}_t^* \sim p(\boldsymbol{\omega}_t \mid \boldsymbol{\omega}_{t-1}, \xi_t)$, driven by unmodeled situational factors $\xi_t$ (e.g., urgency, risk, energy scarcity). In practice, $\xi_t$ is not explicitly modeled but absorbed into the stochastic posterior updates. The agent does not observe $\boldsymbol{\omega}_t^*$ but receives a state stream $s_{t-H+1:t}$ (which may contain observation $o_{t-H+1:t}$ and self-states such as energy and deadline time) over a finite working-memory horizon $H > 1$. Following the Bayesian brain view (Knill & Pouget, 2004; Colombo & Seriès, 2012; Bottemanne, 2025), it maintains beliefs $p(\boldsymbol{\omega}_t \mid s_{t-H+1:t}) \propto p(s_t \mid \boldsymbol{\omega}_t) \cdot p(\boldsymbol{\omega}_t \mid s_{t-H+1:t-1})$ and updates them as new evidence arrives. Note that this generative factorization is an internal perceptual model used by the agent to infer preferences; the actual environment transition kernel $p_{\text{env}}(s_{t+1} \mid s_t, a_t)$ in all our experiments is independent of $\omega_t$ and follows the standard MDP assumption (see Appendix A).

**Latent Representation and Posterior Approximation.** Instead of treating preferences as fixed inputs, we approximate them as distributional latent states that capture both epistemic uncertainty and exploratory variation in preference space. Concretely, we introduce an unconstrained latent vector $\mathbf{z}_t \in \mathbb{R}^d$ with $q_\phi(\mathbf{z}_t \mid s_{t-H+1:t}) = \mathcal{N}(\boldsymbol{\mu}_t, \text{diag}(\boldsymbol{\sigma}_t^2))$, and $\boldsymbol{\omega}_t = \text{softmax}(\mathbf{z}_t)$. This design provides: (i) uncertainty over value preference weight: ambiguous situations produce broader posteriors, explicitly capturing the agent's uncertainty. (ii) exploration in value preference space: instead

of committing to a single trade-off, the agent samples $\mathbf{z}_t$ from the posterior, which slightly perturbs $\boldsymbol{\omega}_t$ and encourages trying nearby preferences before acting. A unit Gaussian prior $p_0(\mathbf{z}) = \mathcal{N}(0, I)$ regularizes the posterior, anchoring values unless evidence strongly suggests change. Detailed implementations are provided in Appendix C.

**Learning objective.** Since the latent preference state $\mathbf{z}_t$ cannot be directly observed, we resort to variational inference and approximate its posterior by $q_\phi(\mathbf{z}_t \mid e_t)$, where $e_t := s_{t-H+1:t}$ denotes the recent state history within a finite working-memory window. We map $\mathbf{z}_t$ to a preference vector on the simplex via a deterministic function $\boldsymbol{\omega}_t = f_\theta(\mathbf{z}_t)$ (e.g., a linear layer followed by a softmax).

This requires specifying how the collected evidence $e_t$ supports different preference configurations. Following bounded-rational choice models (Luce et al., 1959) and the free-energy formulation of Bayesian brain theories (Friston, 2010), we treat the scalarized return under a preference vector as a Boltzmann-rational likelihood of the evidence:

$$p(e_t \mid \mathbf{z}_t) \propto \exp\big(\beta\, U_t(\boldsymbol{\omega}_t; e_t)\big), \quad \beta > 0, \tag{1}$$

where $U_t(\boldsymbol{\omega}_t; e_t) = \langle \boldsymbol{\omega}_t, \vec{G}_t(e_t) \rangle$ represents a scalar utility and $\vec{G}_t(e_t) \in \mathbb{R}^d$ is the vector return estimated by the actor–critic from the history $e_t$.

With an isotropic Gaussian prior $p_0(\mathbf{z}_t) = \mathcal{N}(\mathbf{0}, \mathbf{I})$, Bayes' rule yields the (unnormalized) target posterior

$$p^*(\mathbf{z}_t \mid e_t) \propto p_0(\mathbf{z}_t)\, \exp\big(\beta\, U_t(\boldsymbol{\omega}_t; e_t)\big). \tag{2}$$

We approximate $p^*$ with $q_\phi(\mathbf{z}_t \mid e_t)$ and optimize $q_\phi$ by minimizing $\mathrm{KL}(q_\phi \,\|\, p^*)$, which is equivalent to maximizing the following evidence lower bound (ELBO):

$$\mathcal{L}_{\mathrm{ELBO}} = \beta\, \mathbb{E}_{\mathbf{z}_t \sim q_\phi(\cdot \mid e_t)}\big[U_t(\boldsymbol{\omega}_t; e_t)\big] - \mathrm{KL}\big(q_\phi(\mathbf{z}_t \mid e_t) \,\big\|\, \mathcal{N}(\mathbf{0}, \mathbf{I})\big). \tag{3}$$

This objective balances fit to the current evidence (the utility term) against regularization from the prior, yielding stable yet adaptive updates of preference beliefs. A complete derivation is provided in Appendix A.

## 3.2 Action Selection based on Preference Weights

Humans rarely commit to a single immutable weighting of goals. When facing trade-offs (e.g., morality vs. survival), we may entertain several plausible configurations and act according to the one that seems most promising at the moment. Inspired by this, our agent does not rely on a fixed preference, but rather evaluates a small set of candidates and selects the one that yields the highest predicted utility. To achieve this goal, we employ a preference-conditional policy method, which can adopt policy based on the given preference weights. Specifically, both policy and value functions are conditioned on preferences: $\pi_\theta(a_t \mid s_t, \boldsymbol{\omega}_t), \vec{V}_\theta(s_t, \boldsymbol{\omega}_t) \in \mathbb{R}^d$. Given a preference weight vector, the scalarized value is obtained by $V^{\mathrm{scalar}}(s_t, \boldsymbol{\omega}_t) = \langle \boldsymbol{\omega}_t, \vec{V}_\theta(s_t, \boldsymbol{\omega}_t) \rangle$.

**Envelope Operator.** At each step, $K$ preference candidates are sampled from the appraisal posterior $q_\phi(\cdot \mid s_{t-H+1:t})$, and the one with the highest predicted scalarized value is selected as $\hat{\omega}_t$, e.g.,

$$\hat{\boldsymbol{\omega}}_t = \arg \max_{i \in \{1, \ldots, K\}} \langle \boldsymbol{\omega}_t^{(i)}, \vec{V}_\theta(s_t, \boldsymbol{\omega}_t^{(i)}) \rangle. \tag{4}$$

This is exactly the *envelope operator* (Yang et al., 2019), but executed *on-policy* during action selection, using a single preference-conditioned actor–critic. In addition, we denote by $\boldsymbol{\omega}_t^{\mathrm{pred}}$ preference prediction of the encoder. This $\boldsymbol{\omega}_t^{\mathrm{pred}}$ is used for regularization terms below.

**Vector-valued GAE.** Given vector rewards $\vec{r}_t$, we compute temporal differences for each dimension:

$$\boldsymbol{\delta}_t = \vec{r}_t + \gamma \vec{V}_\theta(s_{t+1}, \hat{\boldsymbol{\omega}}_t) - \vec{V}_\theta(s_t, \hat{\boldsymbol{\omega}}_t), \tag{5}$$

and accumulate vector advantages $\vec{A}_t$ using standard GAE recursion. This ensures temporal credit assignment is handled per-dimension.

**Scalarized advantage for PPO.** The policy update requires a scalar advantage. We project using the same on-policy $\hat{\boldsymbol{\omega}}_t$: $A_t = \langle \hat{\boldsymbol{\omega}}_t, \vec{A}_t \rangle$, $\tilde{A}_t = \mathrm{normalize}(A_t)$, calculating the usual clipped surrogate

$$\mathcal{L}_{\mathrm{PPO}}(\theta) = \mathbb{E}\Big[\min\big(r_t(\theta)\tilde{A}_t,\ \mathrm{clip}(r_t, 1\pm\epsilon)\tilde{A}_t\big)\Big], \tag{6}$$

with $r_t(\theta) = \frac{\pi_\theta(a_t|s_t,\hat{\omega}_t)}{\pi_{\theta \text{old}}(a_t|s_t,\hat{\omega}_t)}$, ensuring on-policy consistency in both action and preference space.

**Dual critic loss.** To stabilize learning process, we combine vector-level supervision with scalarized supervision:

$$\mathcal{L}_{\text{critic}} = \xi \cdot \|\vec{V}_\theta(s_t, \hat{\omega}) - \vec{G}\|_2^2 + (1 - \xi) \cdot (V_\theta^{\text{scalar}}(s_t, \hat{\omega}) - \langle \hat{\omega}, \vec{G} \rangle)^2, \tag{7}$$

where $\vec{G}$ are vectorized returns from vector-GAE.

## 3.3 Stability and Alignment of Preferences

**Preference Alignment.** Intuitively, simply maximizing $U_t(\omega_t) = \langle \omega_t, \vec{G}_t \rangle$ risks degenerate solutions: unstable oscillations of $\omega_t$ or opportunistic "gaming" of temporary fluctuations. Humans avoid this by adjusting preferences smoothly and in line with feasible opportunities. Instead of maximizing utility directly, we introduce two cognitive-inspired regularizers to stabilize the learning process. In detail, to encourage the predicted preference weights $\omega$ aligned with true environment dynamics, we apply a direction alignment loss:

$$\mathcal{L}_{\text{dir}} = \mathbb{E}\left[1 - \frac{\langle \omega_t^{\text{pred}}, \vec{G}_t \rangle}{\|\omega_t^{\text{pred}}\|_2 \, \|\vec{G}_t\|_2}\right], \tag{8}$$

which is applied only when $\|\vec{G}_t\| > 0$. This discourages caring about objectives that are unattainable at the moment.

**Self-consistency.** In addition, to ensure the predicted preference matches the envelope-selected one:

$$\mathcal{L}_{\text{stab}} = \|\omega_t^{\text{pred}} - \hat{\omega}_t\|_2^2. \tag{9}$$

We aim to conduct a posterior instance self-consistency constraint, e.g., if envelope consistently selects a mode, the encoder should directly predict it, reducing policy–preference mismatch.

**Model Optimization.** Together with the ELBO term, the overall training objective of DPI is formulated as

$$\mathcal{L} = \mathcal{L}_{\text{PPO}} + \mathcal{L}_{\text{critic}} - \mathcal{L}_{\text{ELBO}} + \lambda \mathcal{L}_{\text{dir}} + \gamma \mathcal{L}_{\text{stab}}, \tag{10}$$

where $\lambda$ and $\gamma$ are coefficient. The complete training procedure is summarized in Algorithm 1.

## 4 Experiments

In this section, we evaluate our framework along three key aspects: (i) **Q1: Effectiveness**–Does dynamic preference inference improve task performances? (Sec. 4.4) (ii) **Q2: Rationality**–Does the agent adapt its preferences to environmental changes? (Sec. 4.5) (iii) **Q3: Interpretability**–Are the inferred preferences interpretable and semantically aligned? (Sec. 4.6)

## 4.1 Experiment Environments

(i) **Queue.** As is illustrated in Fig. 6a, Queue is a simple but illustrative toy problem, where an agent must decide whether to wait in line or cut ahead to obtain food, balancing two conflicting objectives: energy (survival) and morality (fairness). No fixed weighting suffices: always waiting risks starvation, always cutting erodes morality. Success requires dynamically rebalancing preferences according to context. (ii) **Maze.** We mainly conduct our experiments on Maze environment (as is shown in Fig. 6b), which introduces a 2D navigation task with multiple objectives in pixel space: reaching the goal, meeting deadlines, avoiding hazards, and conserving energy. Random hazard storms and shifting costs introduce non-stationarity, making any static value composition fail. Dynamic value appraisal is essential for progress under varying conditions. (iii) **Continuous control.** We have modified a multi-objective variants of MuJoCo with widely used baselines. See Appendix D.3 for details. Despite differences in modality, these three environments share a structural property: **without dynamic preference adjustment, it is diffucult for the agent to complete the task**.

## 4.2 BASELINES

We compare DPI against a diverse set of representative baselines: (1) **Fixed-Preference MORL (FIXED)**: a strong baseline where the preference vector $\omega$ is fixed to emphasize task-completion objectives (e.g., progress and deadline) (Mossalam et al., 2016), simulating conventional single-objective RL methods that optimize for a scalar reward with secondary penalties treated as regularizers. (2) **Randomized Switching (RS)**: $\omega$ is randomly resampled at runtime, testing whether naive stochastic preference variation can mimic adaptivity. (3) **Heuristic MORL (HEURISTIC)**: hand-crafted preference schedules are applied for different event types and then converted into static $\omega$ settings, serving as a strong rule-based baseline. (4) **Rule-based Envelope Q-learning (ENVELOPE)**: policies conditioned on externally supplied $\omega$ as in a rule-based envelope Q-learning (Yang et al., 2019) with event-dependent preferences (5) **Random Policy (RANDOM)**: a uniformly random agent that samples actions independently at each step, providing a lower bound on performance. This serves as a sanity check to confirm that learned policies achieve meaningful improvement over chance-level behavior. (6) **Dense Oracle**: receives access to the same event signals and additionally uses time / energy information to continuously update preference weights at each timestep (see Appendix .D.2), which provide a practical upper bound under our settings.

## 4.3 EVALUATION METRICS

We report three complementary metrics to comprehensively evaluate overall performance, success rate and adaptation under distribution shifts, respectively. **(i) Mean Episodic Return (MER).** We compute the total return across $N$ episode: $\overline{R} = \frac{1}{N} \sum_{i=1}^{N} \sum_{t=1}^{T} r_t^{(i)}$, where $T$ is the step length and $r_t^{(i)}$ is the scalar reward at step $t$. **(ii) Success Rate (SR).** We measure the fraction of episodes achieving task success: $\text{SR} = \frac{1}{N} \sum_{i=1}^{N} \mathbf{1}\{\psi = 1\}$, where $\psi \in \{0,1\}$ is a task-specific completion flag (e.g., successfully obtaining food in Queue or reaching the goal in Maze). **(iii) Post-Shift Performance (PS@K).** For an environment event occurring at the change point $t^*$, we measure the average return in the first $K$ steps following the change: $\text{PS@}K = \frac{1}{N} \sum_{i=1}^{N} \frac{1}{K} \sum_{t=t^*+1}^{t^*+K} r_t^{(i)}$, which captures adaptation ability after environment contextual shifts. All reported results are averaged over $N = 200$ evaluation episodes for each of 10 random seeds. We report all metrics with the mean and 95% confidence interval (CI) across $N$ episodes: $\text{CI}(\cdot) = 1.96 \cdot \frac{\sigma(\cdot)}{\sqrt{N}}$.

## 4.4 EFFECTIVENESS — DOES DYNAMIC PREFERENCE INFERENCE IMPROVE TASK PERFORMANCE?

| Method | Queue | | Maze | |
|---|---|---|---|---|
| | MER | SR (%) | MER | SR (%) |
| RANDOM | $-24.24 \pm 0.58$ | $17.25 \pm 10.42$ | $-223.55 \pm 10.61$ | $0.00 \pm 0.01$ |
| FIXED | $-4.19 \pm 2.55$ | $10.05 \pm 3.24$ | $16.15 \pm 1.62$ | $1.12 \pm 0.06$ |
| RS | $-4.29 \pm 0.01$ | $11.43 \pm 4.01$ | $-23.66 \pm 4.42$ | $0.01 \pm 0.00$ |
| HEURISTIC | $-1.60 \pm 0.01$ | $10.05 \pm 8.12$ | $-3.65 \pm 0.18$ | $0.00 \pm 0.00$ |
| ENVELOPE | $-3.54 \pm 0.02$ | $25.10 \pm 6.01$ | $10.36 \pm 0.18$ | $0.01 \pm 0.00$ |
| **Ours** | | | | |
| w/ Q-learning | $3.74 \pm 2.30$ | $29.09 \pm 5.33$ | $27.35 \pm 1.25$ | $42.94 \pm 3.72$ |
| w/ PPO | $10.34 \pm 0.02$ | $39.95 \pm 2.75$ | $30.16 \pm 1.22$ | $59.04 \pm 0.10$ |

Table 1: Mean episodic return (MER) and Success rate (SR) across all baselines.

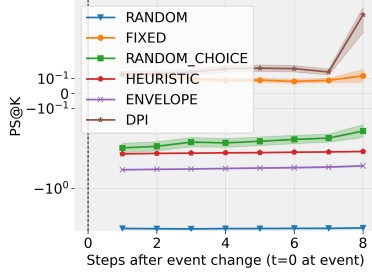

Figure 3: Post-shift performance (PS@K) on Queue environment.

**Toy Environment Result.** We begin with a symbolic Queue environment, a minimal toy setting where the agent must balance survival (energy) against fairness (morality) dynamically. Table 1 shows that our DPI agent substantially improves MER and achieves a 14.85% higher SR than the strongest baseline (ENVELOPE). This demonstrates that dynamically inferred preferences not only improve cumulative performance but also enable the agent to successfully complete the task, whereas fixed or heuristic preferences frequently fail. We additionally analyze the pre- and post-event Pareto fronts over efficiency and fairness, showing that any fixed scalarization necessarily sacrifices performance in at least one regime, whereas DPI adapts its inferred preferences to remain close to the dynamic front (Appendix D.1). Combined with the MER- and SR-PPO (Appendix D.2), this suggests that DPI's gains are not simply due to optimizing a particular scalar metric.

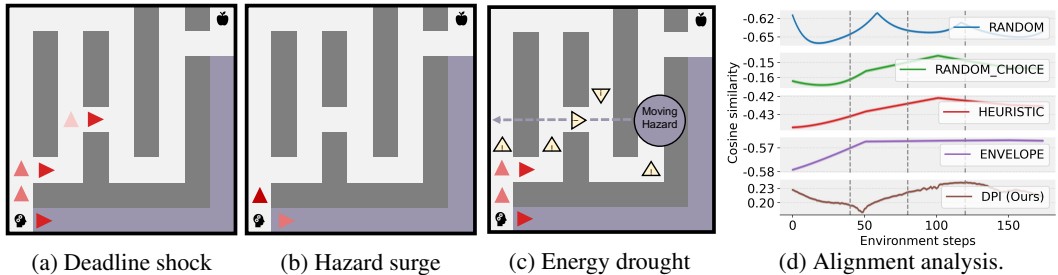

| (a) Deadline shock | (b) Hazard surge | (c) Energy drought | (d) Alignment analysis. |

Figure 4: **Event-aligned trajectories in Maze environment.** After each event, DPI updates its preferences and modifies its behavior in a contextually appropriate way: (a) prioritizes shorter routes under `deadline shock`. (b) exhibits increased avoidance under `hazard surge`. (c) prefers waiting and selecting minimal-cost routes under `energy drought`. Arrows indicate agent motion; shaded regions mark environmental hazards or costs. (d) **Alignment between inferred preferences and reward vectors.** DPI maintains positive cosine similarity and sharply increases alignment after event onsets, whereas baselines remain near zero or negative, indicating that only DPI learns a value representation that tracks task semantics.

**Main result.** Table 1 reports mean episodic return (MER) and success rate (SR) in the Maze environment. RS fails catastrophically, yielding extremely low and highly variable returns. HEURISTIC also performs poorly in this non-stationary setting, indicating that simple hand-designed preference schedules cannot handle the combinatorial diversity of event configurations. ENVELOPE, which is given event-dependent preferences but does not infer them, achieves substantially higher MER, highlighting the importance of conditioning on $\omega$ at deployment. Our Dynamic Preference Inference (DPI) agent achieves the highest overall MER, outperforming ENVELOPE by $+191.1\%$, confirming that online inference over preferences enables more robust long-term behavior under distributional shifts. Classical baselines such as FIXED, HEURISTIC, and ENVELOPE all achieve near-zero SR, even when FIXED attains relatively high MER, showing that they cannot complete the task reliably under all event configurations. DPI attains the highest SR ($59.0\%$), significantly outperforming all ablations and demonstrating its ability to consistently adapt and complete the task under dynamically changing conditions. For completeness, Appendix D.2 further compares DPI against scalarized RL baselines that directly optimize MER or SR (MER-PPO and SR-PPO), as well as a Dense Oracle with privileged access to event signals; DPI still yields superior post-shift performance and success rate in these settings. We further confirm these trends in the modified multi-objective continuous-control setting, where DPI again matches or surpasses strong fixed-preference and oracle baselines (see Appendix D.3).

### 4.5 RATIONALITY–DOES THE AGENT ADAPT ITS PREFERENCES TO ENVIRONMENTAL CHANGES?

To verify that DPI performs meaningful preference adaptation rather than merely exploiting reward structure, we report Post-Shift Performance (PS@K) for $K = 1, \ldots, 8$ in Fig. 3. When events are triggered, both HEURISTIC and the ENVELOPE baselines exhibit persistently low post-shift performance, indicating their inability to adapt to non-stationary dynamics. Interestingly, FIXED attains moderately good PS@K values by greedily pursuing the highest scalarized reward under a static weighting. However, as confirmed in Table 1, this strategy fails to reliably complete the task, demonstrating that no single fixed preference vector is sufficient to handle all event configurations. In contrast, DPI shows rapid recovery after each change point, maintaining high post-shift reward and significantly outperforming all baselines. Together with its $59.04\%$ success rate, these results indicate that DPI is not merely memorizing action sequences, but aligns its internal value preferences with changing environmental demands.

### 4.6 INTERPRETABILITY–ARE THE INFERRED PREFERENCES INTERPRETABLE AND SEMANTICALLY ALIGNED?

**Qualitative evidence: event → preference → behavior.** Fig. 4a, 4b, 4c shows three representative Maze episodes with distinct event types: `deadline shock`, `hazard surge`, and `energy drought`. After each change-point, DPI reweights its preferences in a context-consistent manner and correspondingly switches its behavior—accelerating when deadlines shrink, rerouting when

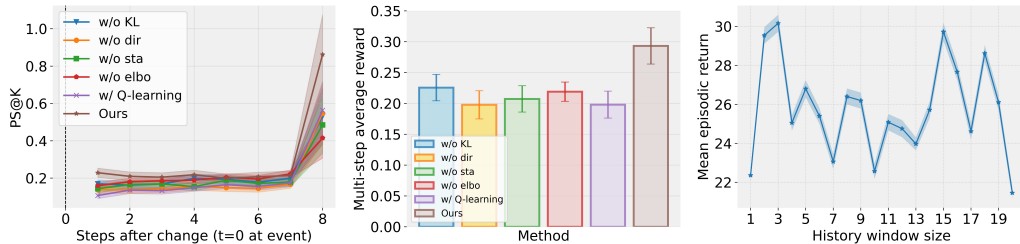

(a) PS@K for each ablated variant.    (b) Multistep average reward.    (c) Different history window size.

Figure 5: **Ablation study results.** (a) Post-Shift Performance (PS@K) curves over the first $K = 8$ steps after each event. (b) Multi-step average PS@K, summarizing short-term recovery into a single metric for each method. (c) Mean episodic return (MER) as a function of history window size $H$. Across all plots, our full DPI agent consistently outperforms ablations, confirming the necessity of KL regularization, directional alignment, and self-consistency, and showing that performance is robust to $H$ beyond a small temporal context.

hazards intensify, and conserving movement under energy scarcity. This event→preference→action chain indicates that DPI is dynamically revising *what matters now* instead of replaying a fixed plan.

**Quantitative alignment with reward structure.** To verify that the inferred preferences track task objectives, we compute the cosine similarity between the inferred preference $\hat{\omega}_t$ and the instantaneous reward vector $\vec{r}_t$: $\mathrm{Align}(t) = \frac{\langle \hat{\omega}_t, \vec{r}_t \rangle}{\|\hat{\omega}_t\|_2 \|\vec{r}_t\|_2}$. As shown in Fig. 4d, DPI maintains consistently positive alignment and exhibits sharp but smooth rises following event onsets, suggesting that its internal value representation reorients toward the most relevant objectives. In contrast, random, heuristic, and envelope baselines remain negative, showing no systematic relation to the reward structure. Together, these results demonstrate that DPI not only adapts its preferences to recover performance, but does so in a semantically interpretable way that reflects the true task demands.

## 4.7 ABLATION STUDY

**Component-wise Ablation.** To isolate the contribution of each design component in DPI, we compare against three variants: (i) **w/o KL**, removing the KL prior term from the ELBO objective; (ii) **w/o dir**, removing the directional alignment constraint between preferences and return gradients; (iii) **w/o sta**, removing the self-consistency regularization that anchors posterior predictions to envelope-selected preferences. We also replace the preference-conditioned actor–critic with a simple Q-learning baseline (**w/ Q-learning**) to test whether a purely value-based method suffices.

Fig. 5a reports Post-Shift Performance (PS@K) over $K = 1, \ldots, 8$ steps after each event, showing how quickly each variant recovers reward after distributional shifts. To provide a single-number summary of short-term recovery, we also report the multistep average PS@K (Fig. 5b), which averages performance across $K = 1 \ldots 8$. Removing any of the three components leads to a measurable performance drop, confirming that KL regularization, directional alignment, and posterior anchoring are all critical for stabilizing preference inference and improving adaptation. Moreover, replacing PPO with Q-learning significantly degrades recovery ability, highlighting the importance of using a preference-conditioned actor–critic for fast on-policy adjustment.

**History Window Size.** We additionally study the effect of the history encoder's receptive field size $H$, varying the number of past observations fed into the preference encoder. Fig. 5c reports the mean episodic return (MER) as a function of $H$ (mean ± 95% CI). Performance degrades significantly for very small history sizes (e.g., $H = 1$), indicating that the model needs sufficient temporal context to infer preferences. However, performance plateaus for moderately large windows ($H \geq 9$), suggesting that our approach is robust to the exact choice of $H$. In all experiments, we choose $H = 3$ as a good trade-off between performance and computational cost.

**Hyperparameter ablations and training stability.** See Appendix D.4 and Appendix D.6 for ablations and learning curves, which show that DPI is robust to the main hyperparameters and trains stably without early collapse.

## 5 CONCLUSION AND FUTURE DIRECTIONS

In this work, we introduced a cognitively inspired framework that abstracts value preference adjustment in dynamic multi-objective environments. We formalized a setting in which preference weights are latent, context-dependent variables that must be inferred online, and proposed DPI, which combines a variational preference inference module with a preference-conditioned policy. Experiments in Queueing, Maze, and multi-objective continuous-control tasks show that DPI enables context-aware preference adaptation and improves performance under event-driven distribution shifts. A key limitation is that our environments remain controlled and simulated rather than open-ended or real-world. Future work will focus on scaling DPI to more realistic 3D embodied and multi-agent settings, and on developing more expressive inference mechanisms for long-horizon and socially coupled preference dynamics.

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

## A  DERIVATION OF THE PREFERENCE OPTIMIZATION OBJECTIVE

**Problem definition.**  At each time step $t$, the agent maintains a latent preference logit vector $\boldsymbol{z}_t \in \mathbb{R}^d$, which is mapped to the probability simplex by

$$\boldsymbol{\omega}_t = f_\theta(\boldsymbol{z}_t) := \operatorname{softmax}(\boldsymbol{z}_t) \in \Delta^{d-1}. \tag{11}$$

Let the evidence at time $t$ be the recent history $e_t := s_{t-H+1:t}$. Given a $d$-dimensional vector return $\vec{G}_t(e_t) \in \mathbb{R}^d$ (estimated by the actor–critic from $e_t$; see Sec. 3.2), we define the scalar utility under preference $\boldsymbol{\omega}_t$ as

$$U_t(\boldsymbol{\omega}_t; e_t) = \langle \boldsymbol{\omega}_t, \vec{G}_t(e_t) \rangle. \tag{12}$$

Our goal is to infer $\boldsymbol{z}_t$ from the evidence $e_t$.

**Assumption 1 (prior and Boltzmann-rational evidence).** (i) Prior: $p_0(\boldsymbol{z}_t) = \mathcal{N}(\mathbf{0}, \boldsymbol{I})$; (ii) Evidence model: following quantal response and free-energy formulations, we posit the *unnormalized* likelihood

$$\log p(e_t \mid \boldsymbol{z}_t) = \beta\, U_t(\boldsymbol{\omega}_t; e_t) + C(e_t), \quad \beta > 0, \tag{13}$$

where $C(e_t)$ does not depend on $\boldsymbol{z}_t$ (hence its gradient vanishes w.r.t. $\phi$).[2]  The corresponding (unnormalized) target posterior is

$$p^*(\boldsymbol{z}_t \mid e_t) \propto p_0(\boldsymbol{z}_t)\, \exp\big(\beta U_t(\boldsymbol{\omega}_t; e_t)\big). \tag{14}$$

**1. Objective: maximize marginal evidence.**  The marginal evidence is

$$p(e_t) = \int p_0(\boldsymbol{z}_t)\, p(e_t \mid \boldsymbol{z}_t)\, d\boldsymbol{z}_t, \tag{15}$$

which is intractable because $\boldsymbol{\omega}_t = f_\theta(\boldsymbol{z}_t) = \operatorname{softmax}(\boldsymbol{z}_t)$ is nonlinear.

**2. Variational family.**  We approximate the posterior by $q_\phi(\boldsymbol{z}_t \mid e_t) = \mathcal{N}(\boldsymbol{\mu}_t, \operatorname{diag}(\boldsymbol{\sigma}_t^2))$, where $(\boldsymbol{\mu}_t, \log \boldsymbol{\sigma}_t)$ are the outputs of the encoder given $e_t$.

**3. KL expansion.**  By Bayes' rule,

$$p^*(\boldsymbol{z}_t \mid e_t) = \frac{p_0(\boldsymbol{z}_t)\, p(e_t \mid \boldsymbol{z}_t)}{p(e_t)}. \tag{16}$$

Thus

$$\mathrm{KL}\big(q_\phi(\boldsymbol{z}_t \mid e_t) \,\|\, p^*(\boldsymbol{z}_t \mid e_t)\big) = \mathbb{E}_{q_\phi}\Big[\log q_\phi(\boldsymbol{z}_t \mid e_t) - \log p_0(\boldsymbol{z}_t) - \log p(e_t \mid \boldsymbol{z}_t)\Big] + \log p(e_t). \tag{17}$$

Rearranging yields

$$\log p(e_t) = \underbrace{\mathbb{E}_{q_\phi}\big[\log p(e_t \mid \boldsymbol{z}_t)\big] - \mathrm{KL}\big(q_\phi(\boldsymbol{z}_t \mid e_t) \,\|\, p_0(\boldsymbol{z}_t)\big)}_{\mathcal{L}_t(\phi)} + \mathrm{KL}\big(q_\phi(\boldsymbol{z}_t \mid e_t) \,\|\, p^*(\boldsymbol{z}_t \mid e_t)\big). \tag{18}$$

**4. ELBO.**  Therefore,

$$\log p(e_t) \geq \mathcal{L}_t(\phi) = \mathbb{E}_{q_\phi(\boldsymbol{z}_t \mid e_t)}\big[\log p(e_t \mid \boldsymbol{z}_t)\big] - \mathrm{KL}\big(q_\phi(\boldsymbol{z}_t \mid e_t) \,\|\, \mathcal{N}(\mathbf{0}, \boldsymbol{I})\big). \tag{19}$$

Substituting the evidence model and dropping the $C(e_t)$ term, we obtain

$$\mathcal{L}_t(\phi) = \beta\, \mathbb{E}_{q_\phi(\boldsymbol{z}_t \mid e_t)}\big[\langle f_\theta(\boldsymbol{z}_t), \vec{G}_t(e_t) \rangle\big] - \mathrm{KL}\big(q_\phi(\boldsymbol{z}_t \mid e_t) \,\|\, \mathcal{N}(\mathbf{0}, \boldsymbol{I})\big). \tag{20}$$

Maximizing $\mathcal{L}_t(\phi)$ is equivalent to minimizing $\mathrm{KL}\big(q_\phi \,\|\, p^*\big)$ and tightens the lower bound on $\log p(e_t)$. Over a window $t = 1, \ldots, T$, the overall preference-inference objective is

$$\max_\phi \sum_{t=1}^T \mathcal{L}_t(\phi) = \sum_{t=1}^T \Big\{ \beta\, \mathbb{E}_{q_\phi(\boldsymbol{z}_t \mid e_t)}\big[\langle f_\theta(\boldsymbol{z}_t), \vec{G}_t(e_t) \rangle\big] - \mathrm{KL}\big(q_\phi(\boldsymbol{z}_t \mid e_t) \,\|\, \mathcal{N}(\mathbf{0}, \boldsymbol{I})\big) \Big\}. \tag{21}$$

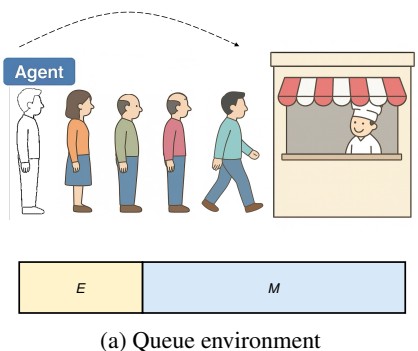
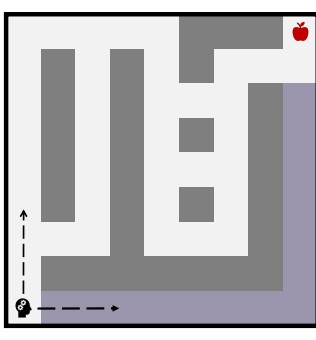

(a) Queue environment                          (b) Maze environment

Figure 6: Two environments used in our experiments. (a) Queue provides a simple but illustrative toy problem. (b) Maze is mainly used in our experiments.

## B    ENVIRONMENT DETAILS

In this section, we will introduce the detailed information of the designed environment in our experiments (as shown in Fig. 6).

### B.1    QUEUE

We design a symbolic queueing environment to capture moral–pragmatic trade-offs such as whether to wait patiently or cut in line under time pressure. The agent waits in a queue to be served and must decide between two discrete actions at each step: WAIT (stay in place) or CUT (jump $k$ positions forward, capped at the front).

**State space.** The agent's state at time $t$ is a compact feature vector

$$s_t = [\text{pos}_t, \text{queue\_len}_t, \text{energy}_t, \text{deadline}_t, \text{recent\_cut}_t, \text{service\_rate}_t],$$

where $\text{pos}_t$ is the agent's current position in the queue (normalized to $[0, 1]$), $\text{queue\_len}_t$ is the remaining queue length, $\text{energy}_t$ is remaining energy, $\text{deadline}_t$ is remaining time budget, $\text{recent\_cut}_t \in \{0, 1\}$ indicates whether the agent just cut the line, and $\text{service\_rate}_t$ is the current service rate (normalized).

**Action space.** The agent chooses from $\mathcal{A} = \{\text{WAIT}, \text{CUT}\}$. Cutting consumes additional energy and may incur a fairness penalty.

**Reward vector.** Each step produces a $d = 5$-dimensional reward vector:

$$\vec{r}_t = \left[ r_t^{\text{progress}}, r_t^{\text{time}}, r_t^{\text{fairness}}, r_t^{\text{energy}}, r_t^{\text{deadline}} \right], \tag{22}$$

where $r_t^{\text{progress}} = +1$ when advancing in the queue or being served, $r_t^{\text{time}} = -1$ per waiting step, $r_t^{\text{fairness}} = -\gamma$ when cutting ahead of others, $r_t^{\text{energy}} = -\lambda$ proportional to energy expenditure, and $r_t^{\text{deadline}} = +\beta$ if served before the deadline.

**Dynamic events.** To induce contextual shifts, we introduce event triggers such as:

(i) Arrival burst: a sudden inflow of new agents increases the queue length;

(ii) Service slowdown: the service rate drops, increasing expected waiting time;

(iii) Energy shock: the agent's remaining energy is reduced, raising the urgency of being served soon. These events create moral–pragmatic conflicts: a patient strategy may lead to starvation if energy runs out, while aggressive cutting risks collapsing the morality score. Thus, no fixed trade-off between fairness and survival can succeed universally, making this environment ideal for testing dynamic preference reweighting.

---

[2]Equivalently, $p(e_t \mid \boldsymbol{z}_t) \propto \exp\left(\beta U_t(\boldsymbol{\omega}_t; e_t)\right)$ with a partition term that is independent of $\boldsymbol{z}_t$.

Table 2: Detailed specifications of environments. Each domain introduces dynamic events that invalidate fixed preferences, requiring agents to adapt their value weights.

| Environment | State Space | Action Space | Reward Vector | Dynamic Events |
|---|---|---|---|---|
| Queue | Symbolic: `pos`, `queue_len`, `energy`, `deadline` | `Wait` / `Cut` | `progress`, `time penalty`, `fairness penalty`, `energy penalty`, `deadline penalty` | Arrival burst; Service slowdown; Energy shock |
| Maze | Pixel: `observation`, `time`, `energy` | `Up` / `Down` / `Left` / `Right` | `progress`, `time penalty`, `hazard penalty`, `energy penalty`, `deadline penalty` | Deadline shock; Hazard surge; Energy drought |

## B.2 Maze

We design a Maze navigation environment with multiple interacting objectives to evaluate adaptive preference inference under dynamic constraints. The agent starts in the bottom-left corner and must reach a goal location in the top-right corner. Each episode terminates upon reaching the goal or exceeding a maximum step budget $T = 200$. At each step the agent observes its current $(x, y)$ position, a global timer normalized to $[0, 1]$, remaining energy, and a binary hazard map indicating nearby dangerous cells. The state is encoded as an image $s_t \in \mathbb{R}^{H \times W \times 3}$, where the three channels are `[observation, time ratio, energy ratio]`, respectively.

**Action space.** The agent chooses from $\mathcal{A} = \{$`UP, DOWN, LEFT, RIGHT`$\}$, moving one cell per step unless blocked by a wall.

**Reward vector.** Each transition produces a $d = 5$-dimensional reward vector:

$$\vec{r}_t = \left[ r_t^{\text{prog}}, r_t^{\text{time}}, r_t^{\text{hazard}}, r_t^{\text{energy}}, r_t^{\text{deadline}} \right],\tag{23}$$

where $r_t^{\text{prog}} = +1$ for moving closer to the goal, $r_t^{\text{time}} = -1$ as a per-step time penalty, $r_t^{\text{hazard}} = -\kappa$ if stepping into a hazardous cell, and $r_t^{\text{energy}} = -\lambda$ proportional to energy consumption, $r_t^{\text{deadline}} = +\beta$ if reached before the deadline. No fixed scalarization is applied; the agent must learn to reweight these objectives.

**Dynamic events.** To induce non-stationarity, we introduce three event types at random steps:

(i) `Deadline shock`: remaining time budget is suddenly shortened by $30\%$, increasing the urgency of reaching the goal.

(ii) `Hazard surge`: a new region of static hazards appears, increasing collision risk along the shortest path.

(iii) `Energy drought`: the agent's energy consumption per step doubles, requiring more conservative movement. Each episode may contain multiple events in sequence, forcing the agent to continuously reappraise what matters most (e.g., prioritize speed when the deadline is tight, or safety when hazards dominate).

Despite their surface differences (as in Table 2), both domains share the same structural property: **without dynamic preference adjustment, the agent rarely succeeds. In our implementations, we empirically observe that no single fixed preference vector over the exposed reward components achieves high MER and SR across all event configurations.** This unified design highlights the necessity of our proposed framework, which equips agents with cognitive-like value reappraisal to remain effective under shifting constraints.

## C  IMPLEMENTATION DETAILS

### C.1  NETWORK ARCHITECTURE

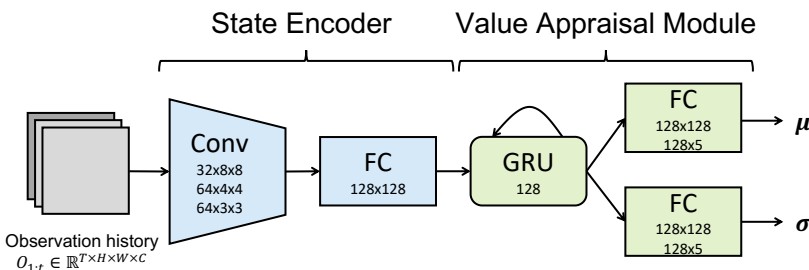

Figure 7: The architecture of our Value Appraisal Module, where the numbers represent channel numbers and kernel size

**Value Appraisal Module.** The network structure of the proposed value appraisal module used in Maze experiments is illustrated in Fig. 7. In detail, the state encoder consists of two convolutional layers and a fully connected layer for encoding observations. Then a GRU network is employed to aggregate historical information, which is connected with two two-layer fully connected network to model the mean and the standard deviation. Each convolutional layer is activated by ReLU, and each fully connected layer is activated by leaky ReLU. In particular, we use two fully connected layers as state encoder for Queue.

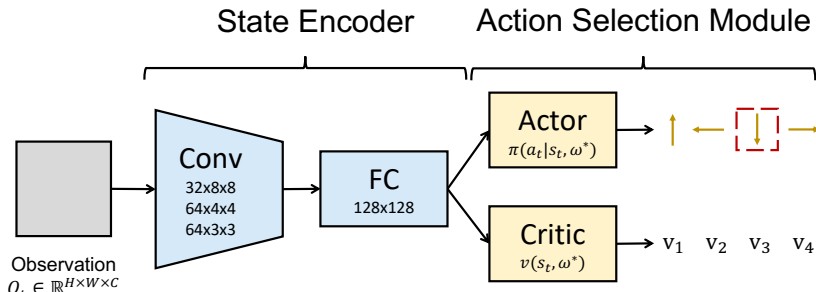

Figure 8: The architecture of our Action Selection Module, where the numbers represent channel numbers and kernel size.

**Action Selection Module.** We employ a value preference conditional actor-critic as our action selection module, as shown in Fig. 8. Specifically, the state encoder is the same as the value appraisal module. The actor and critic are both consists of two fully connected layers activated by leaky ReLU.

### C.2  OPTIMIZATION

The algorithm process for the DPI is shown in Algorithm 1.

The encoder $f_\phi$ outputs $(\mu_t, \log \sigma_t)$ for a diagonal Gaussian $q_\phi(z_t \mid s_{t-H+1:t}) = \mathcal{N}(\mu_t, \mathrm{diag}(\sigma_t^2))$ over the unconstrained latent $z_t$, and preferences are obtained via $\omega(z_t) = \mathrm{softmax}(z_t)$. We estimate the expectation in equation 20 by the reparameterization trick $z_t = \mu_t + \sigma_t \odot \varepsilon, \varepsilon \sim \mathcal{N}(0, I)$, using $1{-}K$ Monte Carlo samples per update. The temperature $\beta$ controls evidence sensitivity (larger $\beta$ gives sharper posteriors. Without loss of generality, we set $\beta = 1$; we keep $\beta$ fixed unless stated otherwise. Vector targets $\vec{G}_t$ are computed per dimension from the critic (see Sec. 3.2), and are treated as constants w.r.t. $\phi$ during preference updates.

---

**Algorithm 1 Dynamic Preference Inference (DPI)** with On-Policy Envelope Actor–Critic

---

**Input:** Policy parameters $\theta$, preference encoder $\phi$
**Repeat** for each training iteration:
   **Collect trajectories:**
     For $t = 1, \ldots, T$:
       Encode recent history $s_{t-H+1:t}$ via GRU $f_\phi$ to obtain mean $\boldsymbol{\mu}_t$, log-variance $\log \boldsymbol{\sigma}_t$:

$$q_\phi(\mathbf{z}_t \mid s_{t-H+1:t}) = \mathcal{N}(\boldsymbol{\mu}_t, \mathrm{diag}(\boldsymbol{\sigma}_t^2)), \quad \boldsymbol{\omega}_t^{(k)} = \mathrm{softmax}(\mathbf{z}_t^{(k)}).$$

       Sample $K$ candidates $\mathbf{z}_t^{(k)}$ and select $\hat{\boldsymbol{\omega}}_t = \arg\max_k \langle \boldsymbol{\omega}_t^{(k)}, \vec{V}_\theta(s_t, \boldsymbol{\omega}_t^{(k)}) \rangle$
       Sample action $a_t \sim \pi_\theta(\cdot \mid s_t, \hat{\boldsymbol{\omega}}_t)$ and step the environment.
   **Optimize:**
     Compute vector returns: $\vec{G}_t$ by multi-objective GAE.
     Compute scalar advantage: $A_t = \langle \hat{\boldsymbol{\omega}}_t, \vec{A}_t \rangle$.
     Compute actor-critic loss: $\mathcal{L}_\pi = \mathcal{L}_{\mathrm{ppo}} + \mathcal{L}_{\mathrm{critic}}$.
     Update $\theta$ by minimizing $\mathcal{L}_\pi$.
     Compute elbo $\mathcal{L}_{\mathrm{ELBO}}$:

$$\mathcal{L}_{\mathrm{ELBO}} = -\mathbb{E}_{\mathbf{z}_t \sim q_\phi}\big[\langle \mathrm{softmax}(\mathbf{z}_t), \vec{G}_t \rangle\big] + \mathrm{KL}\big(q_\phi(\mathbf{z}_t \mid s_{t-H+1:t}) \| \mathcal{N}(\mathbf{0}, \mathbf{I})\big)$$

     Directional alignment regularization: $\mathcal{L}_{\mathrm{dir}} = 1 - \lambda \frac{\langle \boldsymbol{\omega}_t^{\mathrm{pred}}, \vec{G}_t \rangle}{\|\boldsymbol{\omega}_t^{\mathrm{pred}}\|_2 \|\vec{G}_t\|_2}$
     Self-consistency regularization: $\mathcal{L}_{\mathrm{stab}} = \|\boldsymbol{\omega}_t^{\mathrm{pred}} - \hat{\boldsymbol{\omega}}_t\|_2^2$.
     Update $\phi$ by minimizing $\mathcal{L}_{\mathrm{prefer}} = \mathcal{L}_{\mathrm{ELBO}} + \lambda \mathcal{L}_{\mathrm{dir}} + \gamma \mathcal{L}_{\mathrm{sta}}$.

---

### C.3 COMPUTE RESOURCES AND REPRODUCIBILITY

**Compute Resources.** All experiments were conducted on a single workstation equipped with an NVIDIA RTX 4090 GPU (24GB VRAM) and an Intel Core i9-14900K CPU (32 cores, 64GB RAM). This setup allows 6–8 experiments to be run in parallel without resource contention. Training a single DPI agent on Maze for $1.5 \times 10^5$ environment steps takes approximately 15 minutes, and completing all reported experiments across 10 random seeds requires 6 hours in total. Our implementation is based on PyTorch 2.4.1 with CUDA 11.8, and uses Gymnasium 1.0.0 for environment simulation. All code is optimized to run on a single GPU; no distributed training is required.

**Reproducibility.** We fix all random seeds for NumPy, PyTorch, and Gymnasium to ensure reproducibility. Hyperparameters are summarized in Table 3. We report results averaged over 10 independent seeds and present 95% confidence intervals (CI) to account for stochasticity. For ablation studies and sensitivity analyses (e.g., history window size $H$), we sweep parameters in a controlled range and report mean ± CI to ensure robustness.

## D EXTRA EXPERIMENTS

### D.1 PARETO ANALYSIS OF PRE- AND POST-EVENT REGIMES IN QUEUE

To illustrate that the Pareto-optimal trade-offs before and after an event are structurally different, we approximate Pareto fronts in the Queue environment under the pre-event (normal arrival and service rates) and post-event (arrival burst + service slowdown + energy shock) regimes.

For this analysis, we fix a family of linear scalarizations over the vector reward,

$$U(\boldsymbol{\omega}, e) = \langle \boldsymbol{\omega}, \vec{G}(e) \rangle, \tag{24}$$

and sweep $\boldsymbol{\omega}$ over a grid on the simplex. For each $\boldsymbol{\omega}$, we train a policy in the pre-event regime and in the post-event regime, and estimate the corresponding vector returns $\vec{G}_{\mathrm{pre}}(\boldsymbol{\omega})$ and $\vec{G}_{\mathrm{post}}(\boldsymbol{\omega})$ from rollouts. In Fig. 9 we project these vectors onto a 2D objective space: (i) *progress* (higher is better)

Table 3: Hyperparameters used in all experiments. Values are shared across Maze and Queue unless otherwise noted.

| Hyperparameter | Value |
|---|---|
| Learning rate | $2.5 \times 10^{-4}$ |
| Batch size | 1024 transitions per update |
| Discount factor | 0.99 |
| GAE parameter | 0.95 |
| PPO clip ratio ($\epsilon$) | 0.2 |
| Entropy coefficient | 0.01 |
| Value loss coefficient ($\xi$) | 0.5 |
| KL regularization weight ($\alpha_{kl}$) | 0.1 |
| Direction alignment weight ($\lambda$) | 0.1 |
| Self-consistency weight ($\gamma$) | 0.01 |
| Optimizer | Adam |
| Number of epochs per update | 1 |
| History window size ($H$) | 3 (unless otherwise varied in ablation) |
| Preference samples per step ($K$) | 8 |
| Training steps per run | $1.5 \times 10^5$ environment steps |
| Number of seeds | 10 (results reported as mean ± 95% CI) |

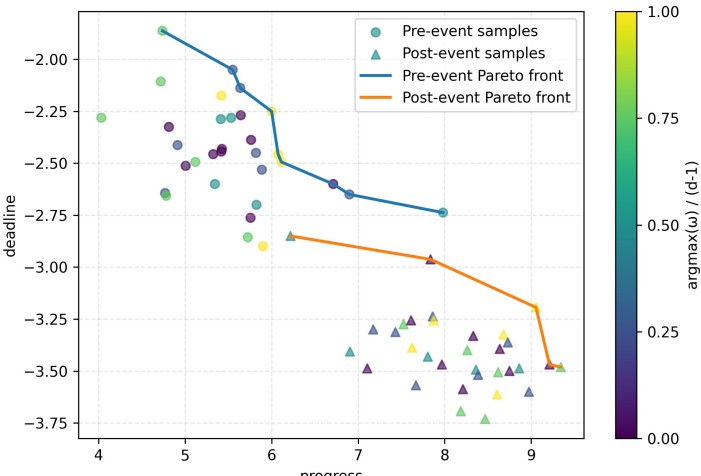

Figure 9: **Queue environment: Pareto fronts before and after the event.** We approximate Pareto fronts in a 2D objective space (progress vs. deadline-related penalty) by training policies with fixed linear scalarizations $\boldsymbol{\omega}$ over the vector reward. Each point is a policy; circles and triangles denote pre- and post-event regimes, respectively, and colors encode the dominant component of $\boldsymbol{\omega}$. The blue curve shows the pre-event Pareto front, which lies predominantly in the upper-left region (smaller deadline penalty with moderate progress), while the orange curve shows the post-event Pareto front, which shifts towards the lower-right (higher progress at the cost of larger deadline penalties). The scalarizations that are near-optimal in the pre-event regime induce strongly suboptimal trade-offs in the post-event regime, and vice versa, indicating that the Pareto-optimal preference configurations before and after the event are largely disjoint. This supports our claim that no single fixed $\boldsymbol{\omega}$ over the exposed reward vector is robust across all event configurations.

on the $x$-axis, and (ii) a *deadline-related penalty* on the $y$-axis (higher on the plot corresponds to smaller penalty / more deadline slack).

Each marker in Fig. 9 corresponds to a policy trained with a fixed scalarization $\boldsymbol{\omega}$: circles denote pre-event policies, triangles denote post-event policies. From these samples we extract the non-dominated points in each regime, which we plot as the pre-event and post-event Pareto fronts.

## D.2 RL BASELINES OPTIMIZING STATIC METRICS AND DENSE ORACLE BASELINE RESULTS

We add three extra baselines in which we tune the scalarization weights separately for MER and SR (MER-PPO and SR-PPO).

which are:

- MER-PPO: PPO with a metric-specific scalar reward $r_t^{\text{MER}} = \langle \omega^{\text{eval}}, \vec{r}_t \rangle$, where the evaluation weight $\omega^{\text{eval}}$ is given to the agent (oracle static scalarization).

- SR-PPO: PPO with a sparse success reward (1 on success, 0 otherwise).

- Dense Oracle: based on PPO that receives access to the same event signals and additionally uses time / energy information to continuously update preference weights at each timestep (e.g., gradually increasing deadline weight as the deadline approaches, increasing hazard weight in proportion to hazard intensity).

We tuned two baselines carefully, and report their performance:

Table 4: Comparison results on Queue and Maze.

| Method | Queue | | Maze | |
|---|---|---|---|---|
| | MER | SR (%) | MER | SR (%) |
| MER-PPO | $15.01 \pm 0.46$ | $0.98 \pm 0.05$ | $85.55 \pm 0.12$ | $0.05 \pm 0.01$ |
| SR-PPO | $-5.64 \pm 5.22$ | $46.91 \pm 0.00$ | $-15.28 \pm 0.96$ | $61.13 \pm 0.05$ |
| Dense Oracle | $14.41 \pm 5.18$ | $49.27 \pm 15.40$ | $40.33 \pm 3.73$ | $62.92 \pm 0.02$ |
| DPI-PPO | $10.34 \pm 0.02$ | $39.95 \pm 2.75$ | $30.16 \pm 1.22$ | $59.04 \pm 0.01$ |

Note that, for MER-PPO and SR-PPO, while these baselines can do well on individual metrics in stationary settings, they struggle to maintain high performance across all metrics under non-stationary event sequences, whereas DPI maintains consistently good MER and SR. In addition, Dense-Oracle baseline can serve as a strong upper bound.

Empirically, PPO-MER achieves competitive mean episodic return, but still suffers from low success rate, confirming that single fixed scalarization can hardly cope with all event configurations. Our DPI agent maintains comparable MER while substantially improving SR, and additionally provides interpretable, event-aligned preference trajectories (as shown in Fig. 4). This supports our claim that explicit dynamic preference inference offers benefits beyond standard single-objective RL. In addition, designing a Markovian reward that directly optimizes PS@K would require explicit access to change points and the evaluation horizon $K$, which we intentionally do not assume. We therefore treat PS@K as an evaluation-only metric and focus on MER/SR for single-objective RL baselines.

## D.3 CONTINUOUS CONTROL ENVIRONMENT EXPERIMENTS

To demonstrate the scalability of the proposed method, we modified and implemented a continuous environment based on multi-objective mujoco. The experiment result is reported in Table 5.

We further construct a continuous-control benchmark by extending the multi-objective HalfCheetah environment from `mo-gymnasium` (`mo-halfcheetah-v5`) with dynamic events and resource constraints. The agent controls a planar cheetah to run forward while trading off forward progress, control effort, and energy consumption under a stochastic schedule of events. Compared to the grid-based Maze in the main text, this environment probes preference inference in a higher-dimensional, continuous state–action space.

**State and observation.** Let $o_t \in \mathbb{R}^{d_o}$ denote the standard MuJoCo observation of HalfCheetah at time $t$ (joint positions/velocities, torso state, etc.). To provide short-term temporal context, we expose to the agent a stacked history of length $H$:

$$s_t = \left[ o_{t-H+1}, \ldots, o_t \right] \in \mathbb{R}^{H \times d_o}, \tag{25}$$

with $H = 8$ in all experiments. Internally, the environment also maintains a remaining deadline $D_t \in \mathbb{N}$ and an energy budget $E_t \in \mathbb{R}_+$, which govern termination but are not directly observed by the agent (making the task partially observable with respect to constraints). We set a maximum horizon $T_{\max} = 200$, an initial deadline $D_0 = 200$, and an initial energy budget $E_0 = 100$.

**Action space.**  The action $a_t \in \mathbb{R}^{d_a}$ is a continuous torque vector applied to the actuated joints, identical to the standard HalfCheetah control interface. We use the same box-constrained action space as `mo-halfcheetah-v5`.

**Reward vector and constraints.**  At each step the underlying `mo-halfcheetah-v5` environment returns a vector reward

$$\tilde{\mathbf{r}}_t = [\tilde{r}_t^{\text{speed}}, \tilde{r}_t^{\text{ctrl}}], \tag{26}$$

where $\tilde{r}_t^{\text{speed}}$ is the forward speed reward (proportional to the $x$-velocity of the torso) and $\tilde{r}_t^{\text{ctrl}}$ encodes a control-cost penalty.

We construct a $d = 3$ dimensional reward vector

$$\vec{r}_t = \left[ r_t^{\text{speed}}, \ r_t^{\text{ctrl}}, \ r_t^{\text{energy}} \right], \tag{27}$$

with

$$\begin{aligned}
r_t^{\text{speed}} &= \alpha_t^{\text{speed}} \tilde{r}_t^{\text{speed}}, \\
r_t^{\text{ctrl}} &= -\alpha_t^{\text{ctrl}} \left| \tilde{r}_t^{\text{ctrl}} \right|, \\
r_t^{\text{energy}} &= -\eta_t \, \|a_t\|_2 \ + \ r_t^{\text{deadline}}.
\end{aligned} \tag{28}$$

Here $\alpha_t^{\text{speed}}, \alpha_t^{\text{ctrl}}, \eta_t > 0$ are time-varying scales that may change when events occur (see below). The energy budget is updated as

$$E_{t+1} = E_t - \eta_t \, \|a_t\|_2, \tag{29}$$

and the deadline counts down as $D_{t+1} = D_t - 1$.

The deadline component $r_t^{\text{deadline}}$ is only non-zero at termination. If the episode terminates due to reaching a goal state before exhausting the deadline or energy budget, we grant a positive bonus $+\beta$; if it terminates due to deadline expiration, we assign a penalty $-\beta$; and if it terminates due to energy exhaustion, we add an extra negative penalty on $r_t^{\text{energy}}$. In our implementation we use $\beta = 10$. The environment terminates when either (i) the underlying MuJoCo simulator signals failure, (ii) $D_t \leq 0$ (deadline reached), (iii) $E_t \leq 0$ (energy exhausted), or (iv) the maximum horizon $T_{\max}$ is reached (in which case we treat the episode as truncated). No fixed scalarization is hard-coded into the environment; as in Maze, the agent must adaptively reweight the components of $\vec{r}_t$.

**Dynamic events.**  To induce non-stationary trade-offs, we introduce three types of exogenous events that modify constraints and reward scales:

`Deadline shock`: the remaining time budget is suddenly shortened by a fraction. Concretely, at event time $t$, we sample a shrink ratio $\rho \in (0, 1)$ (default $\rho = 0.5$) and update

$$D_t \leftarrow \max\{1, \ D_t - \lceil \rho D_t \rceil\}, \tag{30}$$

increasing the urgency of progress.

`Speed surge`: the importance of forward speed is increased by multiplying its scale:

$$\alpha_t^{\text{speed}} \leftarrow c_{\text{speed}} \, \alpha_t^{\text{speed}}, \tag{31}$$

with default multiplier $c_{\text{speed}} = 1.5$. This models scenarios where performance pressure (e.g., a tighter external requirement on forward velocity) suddenly rises.

`Energy drought`: the per-step energy cost increases, making actions more expensive:

$$\eta_t \leftarrow c_{\text{energy}} \, \eta_t, \tag{32}$$

with default multiplier $c_{\text{energy}} = 2.0$. This mimics a sudden reduction in available power or efficiency, pushing the agent to adopt more conservative control.

Table 5: Comparison results on the modified continuous control environment. To handle a continuous action space, we adapt the rule-based ENVELOPE Q-learning baseline to a PPO actor–critic implementation (ENVELOPE-PPO). Dense Oracle is given privileged access to dynamic event signals and hand-crafted rules for updating the preference weights, whereas DPI-PPO (ours) only observes the standard environment state and must infer preferences purely from interaction.

| Method | MER | SR(%) |
|--------|-----|-------|
| ENVELOPE-PPO | $13.52 \pm 9.89$ | $53.28 \pm 12.00$ |
| Dense Oracle | $66.95 \pm 5.78$ | $99.98 \pm 0.01$ |
| DPI-PPO (Ours) | $42.10 \pm 6.25$ | $81.00 \pm 1.00$ |

### D.4 HYPERPARAMETER SENSITIVITY ANALYSIS

To evaluate how sensitive DPI is to the auxiliary loss coefficients, we conduct a experiment on the QUEUE environment, varying the stability weight $\gamma$ and the directional-alignment weight $\lambda$ in Eq. (10) while keeping all other hyperparameters fixed. For each setting, we train the agent with 10 random seeds and report the mean episodic return (MER) and success rate (SR) over 200 evaluation episodes in Table 6. Overall, DPI is fairly robust to moderate changes of $\gamma$ and $\lambda$: performance under all tested configurations stays within one standard deviation of the default setting, and only very small or very large coefficients lead to a mild degradation in SR.

Table 6: Hyperparameter sensitivity analysis. Here $\gamma$ and $\lambda$ are reported as relative multipliers around the default $(\gamma_0, \lambda_0)$ used in the main experiments (i.e., 0.5 means $0.5\gamma_0$).

| $\gamma$ | $\lambda$ | MER | SR(%) |
|------|------|-----|-------|
| 0.5 | 1.0 | $10.84 \pm 1.33$ | $38.57 \pm 2.10$ |
| 1.0 | 1.0 | $10.34 \pm 0.02$ | $39.95 \pm 2.75$ |
| 2.0 | 1.0 | $9.55 \pm 0.62$ | $35.85 \pm 1.02$ |
| 1.0 | 0.5 | $9.70 \pm 0.78$ | $30.43 \pm 3.27$ |
| 1.0 | 2.0 | $8.85 \pm 1.62$ | $36.93 \pm 0.46$ |

### D.5 DISCUSSION ON COLD-START DYNAMICS

In our implementation, early training is stabilized by two mechanisms.

**(i) Gaussian prior and KL regularization.** Because $q_\phi(z_t \mid \cdot)$ is regularized toward $\mathcal{N}(0, I)$, the induced preference weights $\omega_t = \mathrm{softmax}(z_t)$ are initially close to a high-entropy, nearly uniform distribution over objectives. This prevents the appraisal module from committing to extreme trade-offs when the critic is still inaccurate.

**(ii) Envelope selection with on-policy actor–critic.** Even when the multi-objective returns $\vec{G}_t$ are noisy, the envelope operator is applied over multiple $\omega_t$ samples drawn from a broad posterior, and the PPO update is on-policy with respect to the selected $\omega_t$. In practice, this behaves similarly to a standard multi-objective actor–critic with stochastic exploration in preference space, rather than a brittle deterministic scheduler.

Empirically, we do not observe systematic early mode collapse of preferences. Instead, the entropy of $\omega_t$ gradually decreases as the critic becomes more informative, consistent with a smooth transition from exploratory to more specialized preference configurations.

### D.6 LEARNING CURVES

To assess the training stability of DPI, we run 10 independent training runs on the Queue environment with different random seeds, using the default hyperparameters reported in Table 3. For each run, we log performance throughout training as follows:

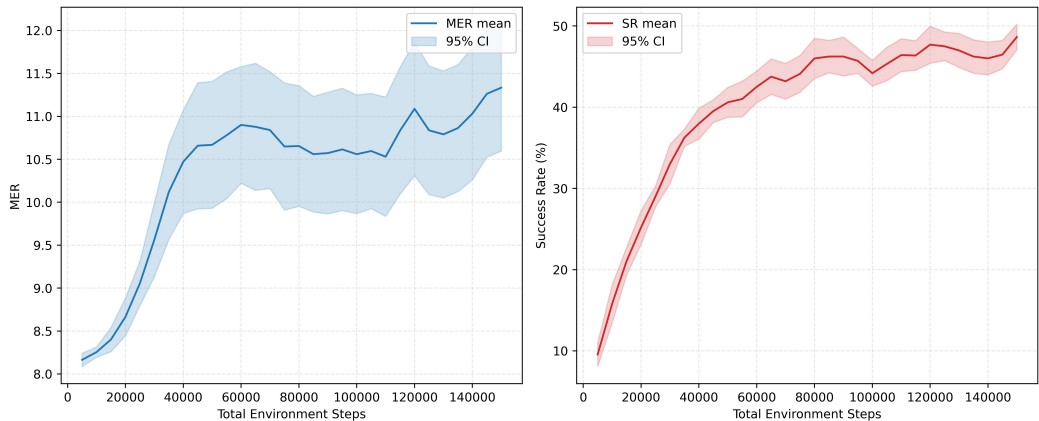

Figure 10: Mean episodic return (MER, left) and success rate (SR, right) as a function of environment steps. Solid lines show the mean over 10 random seeds and shaded regions denote the 95% confidence interval. DPI-PPO exhibits smooth and monotone improvement without divergence or collapse.

- After every 5,000 environment interaction steps, we fix the current policy and derive a greedy evaluation policy (breaking action ties uniformly at random).
- We then evaluate this policy on 200 episodes without exploration noise and compute the mean episodic return (MER) and success rate (SR) for that evaluation point.

For each evaluation step, we aggregate MER and SR across seeds and report the mean and the 95% confidence interval, which are plotted in Fig. 10. As shown in the figure, both MER and SR improve smoothly over time and eventually saturate, without divergence, collapse, or large oscillations. These observations indicate that the combined ELBO-based preference inference and PPO optimization yield stable training dynamics.

