# OpenReview forum: "Learning What Matters Now: Dynamic Preference Inference under Contextual Shifts"
_ICLR.cc/2026/Conference — ICLR 2026 Poster_

### Official Review · Reviewer_9DQE · 2025-10-28

**Soundness:** 3
**Presentation:** 3
**Contribution:** 3
**Rating:** 6
**Confidence:** 3

**Summary:**

A cognitively inspired RL framework is proposed for dynamically inferring agent preferences in non-stationary, multi-objective environments. The method jointly trains preference inference and control via variational Bayesian optimization, achieving interpretable and adaptive behavior. Experiments in two environments demonstrate the effectivenss of the proposed method.

**Strengths:**

1. The central idea of treating value preferences as latent, context-sensitive cognitive states inferred through Bayesian variational methods is novel and interesting.

2. The manuscript is clearly organized and flows logically from motivation to methodology to experiments.

3. The experimental design is good, featuring comprehensive analyses and ablation studies that offer valuable insights into the model’s behavior.

**Weaknesses:**

1. The experiments are confined to relatively small, synthetic domains (toy symbolic and 2D Maze). Although cognitively illustrative, these environments may not fully demonstrate the scalability or generalizability of the proposed DPI framework. More complicated continous environments or real-world cases that better reflect the changing preferences are needed.

2. Although the paper introduces a principled ELBO-based training objective with regularization terms, it lacks sufficient analysis of training stability, including convergence behavior and sensitivity to hyperparameters.

**Questions:**

1. In Lines 93–94, the definition of $\omega_t$ is unclear. Please clarify its meaning and justify why it is denoted in bold.

2. The choice of a unit Gaussian prior lacks sufficient motivation. It would be helpful to provide a rationale or empirical justification for this assumption.

3. In Line 203, the derivation of the presented formula is not fully explained. Additional details or intermediate steps would improve clarity.

4. In the Maze environment, the authors claim that “without dynamic preference adjustment, the agent cannot succeed.” Please elaborate on how this statement was verified or implemented in practice.

5. In Table 1, the success rate (SR) should also be reported with the mean and 95% confidence interval (CI) across $N$ episodes, consistent with the other metrics.

---

> ### Author Response · Authors · 2025-11-21
> **Response to Reviewer 9DQE[1/2]**
>
> We sincerely thank the reviewer for the careful evaluation and the positive assessment of our work. We are very encouraged by your comments that (i) treating preference weights as latent, context-sensitive cognitive states is novel and interesting, (ii) the manuscript is clearly organized, and (iii) the experimental design and ablations are informative. We also appreciate your constructive suggestions on limitations and clarification points. Below we address your comments in detail.
>
> >   **W1**: Limited to synthetic domains
>
> **R1:** We agree that the current experiments are confined to symbolic queue and 2D maze environments, which are cognitively illustrative but do not fully demonstrate scalability to more complex, high-dimensional tasks. Following your suggestion, we modified and implemented a continuous environment based on multi-objective mujoco. Details of the environment are provided in `Appendix D.3`. We report Mean Episode Return (MER, higher is better) and Success Rate (SR) of the Rule-based Envelope PPO (ENVELOPE-PPO), a Dense Oracle PPO and our method's performance:
>
> | Method       | MER             | SR(%)            |
> | ------------ | --------------- | ---------------- |
> | ENVELOPE     | $13.52\pm 8.72$ | $53.28\pm 12.00$ |
> | Dense Oracle | $66.95\pm 5.78$ | $99.98\pm 0.01$  |
> | DPI-PPO      | $42.10\pm 9.89$ | $81.99\pm 8.15$  |
>
> Due to space, we keep the empirical scope of this initial work to two synthetic domains in the main text, but we agree that extending DPI to more realistic settings is an important direction for future work and will highlight this more clearly.
>
> >   **W2**:  Stability and sensitivity analysis
>
> **R2:** Thanks for this valuable suggestion. We also agree that the current presentation does not sufficiently discuss stability and sensitivity to hyperparameters. Here we conduct a short **hyperparameter sensitivity experiment**:
>
> | $\gamma$ | $\lambda$ | MER              | SR(%)            |
> | -------- | --------- | ---------------- | ---------------- |
> | 0.5      | 1.0       | $10.84\pm 1.33$  | $38.57 \pm 2.10$ |
> | 1.0      | 1.0       | $10.34 \pm 0.02$ | $39.95 \pm 2.75$ |
> | 2.0      | 1.0       | $9.55 \pm 0.62$  | $35.85 \pm 1.02$ |
> | 1.0      | 0.5       | $9.70 \pm 0.78$  | $30.43 \pm 3.27$ |
> | 1.0      | 2.0       | $8.85 \pm 1.62$  | $36.93\pm 0.46$  |
>
> The results are based on the Queue environment. Here $\gamma$ and $\lambda$ are reported as relative multipliers around the default $(\gamma_0, \lambda_0)$ used in the main experiments (i.e., 0.5 means $0.5\gamma_0$).
>
> Across these settings, we did not observe obvious catastrophic degradation, suggesting that DPI is reasonably robust to moderate changes of $\lambda$ and $\gamma$. These results have been added to the `Appendix D.4`.
>
> We hope these additions will make the stability properties of DPI more transparent.
>
> >   **Q1:** Definition and bold notation of  $\boldsymbol{\omega}_t$
>
> **R3:**  Thank you for pointing out that the definition of $\boldsymbol{\omega}_t$ was not sufficiently clear. In the revised manuscript, we have:
>
> -   Clearly defined $\boldsymbol{\omega}_t \in \Delta^{d-1}$ as the **vector of preference weights over the** d **reward dimensions** at time t;
> -   Explicitly stated that the boldface indicates that $\boldsymbol{\omega}_t$ is a vector; and
> -   Moved this definition earlier in Section 3 to ensure consistent usage.
>
> >   **Q2:** Motivation for the unit Gaussian prior
>
> **R4:** We chose the unit Gaussian prior $p_0(z_t) = \mathcal N(\mathbf{0},\mathbf{I})$ in latent space for two reasons:
>
> 1.  **Neutrality and symmetry.** It does not a priori favor any particular objective dimension, which is desirable when we have no strong prior belief that one component should dominate. Combined with the softmax mapping $\omega_t = f_\theta(z_t)$, this yields initial preferences that are close to uniform.
> 2.  **Regularization.** The KL term encourages the posterior $q_\phi(z_t \mid s_{t-H+1:t})$ to stay near this neutral prior when evidence is weak, preventing extreme preferences early in training and thereby improving stability.
>
> We will add this rationale to Section 3 and note that more structured priors (e.g., sparse, hierarchical, or with explicit temporal correlations) are an interesting avenue for future work.

---

> ### Author Response · Authors · 2025-11-21
> **Response to Reviewer 9DQE[2/2]**
>
> > **Q3:** Derivation of the formula around Line 203
>
> **R5:** We appreciate that the current derivation is too compressed. In the revision:
>
> -   We have expanded the derivation in `Appendix A`. We explicitly introduce $e_t := s_{t-H+1:t}$ as the evidence and start from the Boltzmann-rational likelihood
>     $$
>     p(e_t \mid z_t) \propto \exp\big(\beta\, U_t(\omega_t; e_t)\big)
>     $$
>     combine it with the Gaussian prior to obtain the unnormalized posterior
>
>     $$
>     p^\ast(z_t \mid e_t) \propto p_0(z_t)\exp\big(\beta U_t(\omega_t; e_t)\big),
>     $$
>     and then show that maximizing the ELBO is equivalent to minimizing $\mathrm{KL}(q_\phi \| p^\ast)$;
>
> -   We will add a short explanatory paragraph in the main text that references the appendix and makes the probabilistic structure of Equations (1–3) explicit.
>
> This should make it easier for readers to follow the Bayesian interpretation of the learning objective.
>
> >   Q4: “Without dynamic preference adjustment, the agent cannot succeed” in Maze
>
> **R6:** You are right that this statement needs to be justified more carefully. What we meant is that, as reported in `Table 1`, **in our particular Maze configuration**, we empirically observed that a wide range of **static scalarizations** (fixed preference weights) failed to achieve high success rate across the diverse event sequences we sample (deadline shocks, hazard surges, energy droughts). DPI, in contrast, adapts its latent preferences to these changes and maintains high performance.
>
> To clarify this, we have rephrased the sentence in the main text to avoid sounding like a universal claim. Instead, we will state that “in our instantiated Maze setting, static preferences perform poorly under the range of non-stationary event sequences we consider, which motivates dynamic preference adjustment.”
>
> >   Q5. Reporting SR with mean and 95% CI
>
> **R7:** Thanks for this suggestion. In the revised version of manuscript, we have report **SR as mean ± 95% confidence interval** over runs, consistent with how we already report MER and PS@K;
>
> Once again, we thank you for the constructive comments and for highlighting both the strengths and limitations of our work. Hope that the above clarifications resolve your concerns. We sincerely welcome any further questions or suggestions you may have.

---

### Official Review · Reviewer_6Kvv · 2025-11-01

**Soundness:** 2
**Presentation:** 3
**Contribution:** 3
**Rating:** 4
**Confidence:** 4

**Summary:**

The paper addresses the challenge of Multi-Objective Reinforcement Learning in dynamic environments where preference weights are not fixed or explicitly given, but are instead latent and subject to change based on contextual shifts. The authors propose a cognitive-inspired framework called Dynamic Preference Inference (DPI). DPI consists of two main components: a Value Appraisal Module that uses variational inference to maintain a posterior distribution over latent preference weights based on recent history, and an Action Selection Module that employs a preference-conditioned actor-critic to execute policies based on sampled preferences. The framework is validated on two synthetic environments (Queue and Maze), showing that it can adapt to sudden changes in task dynamics (e.g., "deadline shocks" or "hazard surges") better than static or heuristic baselines.

**Strengths:**

- Novel Problem Formulation: The paper identifies and formalizes a significant gap in current MORL research: treating preference weights as latent and dynamic states that must be inferred online, rather than static inputs. This is well-motivated by cognitive theories of human decision-making.

- Principled Framework: The DPI method offers a principled probabilistic approach, synthesizing variational inference for preference estimation (using ELBO) with established preference-conditioned RL techniques (like the envelope operator).

- Demonstrated Adaptivity: Empirical results on the test environments clearly demonstrate the method's ability to recover performance (measured by Post-Shift Performance, PS@K) rapidly after unobserved environmental shifts, significantly outperforming fixed-preference baselines.

- Interpretability: The inclusion of alignment analysis (cosine similarity between inferred preferences and instantaneous reward vectors) provides evidence that the learned latent preferences meaningfully track shifting task semantics.

**Weaknesses:**

- Limited Experimental Scope: The evaluation is restricted to two relatively simple synthetic environments: a symbolic queue task and a 2D grid-world. While illustrative, these do not fully demonstrate the method's scalability to complex, high-dimensional, or continuous control problems often seen in practical MORL settings.

- Potentially Weak Oracle Baseline: The ENVELOPE baseline, described as having "oracle access to event-dependent preference weights" , performs surprisingly poorly (e.g., only 0.01% success rate in Maze ). This baseline appears to use sparse, static weights triggered by events. A true "oracle" in such highly dynamic environments should likely use dense preference updates at every timestep (e.g., exponentially increasing urgency as a deadline approaches, not just a flat jump in weight when a "shock" occurs). The large performance gap might simply reflect poorly tuned static weights for the baseline rather than the absolute necessity of inference.

- Complexity of Objective: The total training objective is a composition of many terms: PPO loss, dual critic loss, ELBO, directional alignment ($\mathcal{L}\_{dir}$), and self-consistency ($\mathcal{L}\_{stab}$). The ablation study confirms standard components alone (e.g., w/o $\mathcal{L}\_{dir}$ or $\mathcal{L}\_{stab}$) perform significantly worse, suggesting the method might be highly sensitive to the hyperparameters balancing these diverse loss terms ($\lambda, \gamma, \xi, \beta$).

- Dependence on Return Vectors for Alignment: The directional alignment loss relies on vector returns $\vec{G}\_t$ estimated by the critic. In environments with very sparse rewards or difficult exploration challenges, these return estimates might be highly inaccurate early in training, potentially leading to poor preference inference that is hard to recover from.

**Questions:**

1. Dense Oracle Baseline: Could you compare DPI against a "Dense Oracle" baseline that has access to analytically derived optimal preferences at every timestep (e.g., analytically derived from exact remaining time and energy)? This would provide a stronger upper bound to gauge the true effectiveness of DPI's inference module compared to standard envelope methods.

2. Hyperparameter Sensitivity: Given the composite nature of the loss function, how sensitive is the model's performance to variations in the auxiliary loss coefficients $\lambda$ (alignment) and $\gamma$ (stability)?

3. Cold-Start Dynamics: How does DPI behave early in training when the critic's vector return estimates $\vec{G}_t$ are likely noisy or uninformative? Does this lead to early collapse into sub-optimal preference modes?

4. Theoretical vs. Practical Gap: The generative assumption is $\omega\_{t}^{*}\sim p(\omega\_{t} | \omega\_{t-1},\xi\_{t})$, but the implementation uses a static prior $p\_{0}(z)=\mathcal{N}(0,I)$. While the recurrent encoder handles history, did you experiment with strictly enforcing the temporal dependency in the ELBO (e.g., a learned transition prior)?

---

> ### Author Response · Authors · 2025-11-21
> **Response to Reviewer 6Kvv[1/2]**
>
> We sincerely thank the reviewer for the detailed and constructive feedback, and for highlighting both the novelty of the problem formulation and the strengths of our probabilistic framework and interpretability analysis. Below we address your concerns and respond to your questions.
>
> >   **W1**: Scope of the experimental evaluation
>
> **R1:** Thank you for pointing out the limitations of the lack of continuous environments in our experiments. Following your suggestion, we modified and implemented a continuous environment based on multi-objective mujoco. Details of the environment are provided in `Appendix D.3`. We fully share your interest in scaling the approach to more complex continuous-control settings. Here we report Mean Episode Return (MER, higher is better) and Success Rate (SR) of the Rule-based Envelope PPO (ENVELOPE-PPO) and our method's performance:
>
> | Method         | MER             | SR(%)            |
> | -------------- | --------------- | ---------------- |
> | ENVELOPE-PPO   | $13.52\pm 9.89$ | $53.28\pm 12.00$ |
> | Dense Oracle   | $66.95\pm 5.78$ | $99.98\pm 0.01$  |
> | DPI-PPO (Ours) | $42.10\pm 6.25$ | $81.00 \pm 1.00$ |
>
> We have included these new results in the revised manuscript (`Appendix D.3`).
>
> >   **W2&Q1**: Dense Oracle Baseline
>
> **R2:** We appreciate your careful reading and agree that our current ENVELOPE baseline is closer to an **event-triggered rule-based method** (with piecewise-static weights) than a fully dense oracle, and that this may underestimate what an oracle with access to ground-truth preference dynamics could do. We have revised the text to avoid calling this an “oracle” baseline and instead describe it explicitly as a rule-based envelope method with event-dependent preferences. This should prevent over-interpretation of its performance as a true upper bound.
>
> Here we:
>
> -   reimplement a **Dense Oracle** baseline based on PPO that receives access to the same event signals and additionally uses **time / energy** information to continuously update preference weights at each timestep (e.g., gradually increasing deadline weight as the deadline approaches, increasing hazard weight in proportion to hazard intensity). In contrast, DPI must learn such preference dynamics purely from interaction, without access to these oracle rules.
> -   Use this baseline both in Queue and Maze to provide a practical upper bound under our implementation.
>
>
> | Method        | Queue          |                 | Maze           |                 |
> | ------------- | -------------- | --------------- | -------------- | --------------- |
> |               | **MER**        | **SR(%)**       | **MER**        | **SR**          |
> | Dense Oracle  | $14.41\pm5.18$ | $49.27\pm15.40$ | $40.33\pm3.73$ | $62.92\pm 0.02$ |
> | DPI-PPO(Ours) | $10.34\pm0.02$ | $39.95\pm 2.75$ | $30.16\pm1.22$ | $59.04\pm 0.01$ |
>
> Thanks for this valuable suggestion. We have added these results to `Appendix D.2` and revised the description of the Rule-based ENVELOPE baseline in the main text accordingly.
>
> >   **W3&Q2**: Hyperparameter sensitivity (λ, γ)
>
> **R3:** We agree that understanding sensitivity to the auxiliary loss coefficients is important. Here we conduct a short **hyperparameter sensitivity experiment**:
>
> | $\gamma$ | $\lambda$ | MER              | SR(%)            |
> | -------- | --------- | ---------------- | ---------------- |
> | 0.5      | 1.0       | $10.84\pm 1.33$  | $38.57 \pm 2.10$ |
> | 1.0      | 1.0       | $10.34 \pm 0.02$ | $39.95 \pm 2.75$ |
> | 2.0      | 1.0       | $9.55 \pm 0.62$  | $35.85 \pm 1.02$ |
> | 1.0      | 0.5       | $9.70 \pm 0.78$  | $30.43 \pm 3.27$ |
> | 1.0      | 2.0       | $8.85 \pm 1.62$  | $36.93\pm 0.46$  |
>
> Here $\gamma$ and $\lambda$ are reported as relative multipliers around the default $(\gamma_0, \lambda_0)$ used in the main experiments (i.e., 0.5 means $0.5\gamma_0$).
>
> Across these settings, we did not observe obvious catastrophic degradation, suggesting that DPI is reasonably robust to moderate changes of $\lambda$ and $\gamma$. These results have been added to the `Appendix D.4`.

---

> ### Author Response · Authors · 2025-11-21
> **Response to Reviewer 6Kvv[2/2]**
>
> >   **Q3**: Cold-start dynamics
>
> **R4:** This is an important point. In our implementation, early training is stabilized by two mechanisms:
>
> 1.  **Gaussian prior and KL term.**
>
>     Because $q_\phi(z_t | \cdot)$ is regularized towards $N(0, I)$, the corresponding $\boldsymbol\omega_t = \operatorname{softmax}(\boldsymbol z_t)$ is initially close to a **high-entropy, nearly uniform** preference over objectives. This prevents the appraisal module from over-committing to any extreme trade-off when the critic is still inaccurate.
>
> 2.  **Envelope selection with on-policy actor–critic.**
>
>     Even when $\vec G_t$ is noisy, the envelope operator is applied over multiple $\omega_t$ samples drawn from a broad posterior, and the PPO update is on-policy w.r.t. the selected $\boldsymbol \omega_t$. In practice, this behaves similarly to a standard multi-objective actor–critic with added stochastic exploration in preference space, rather than a brittle deterministic scheduler.
>
> Empirically, we did not observe systematic early “mode collapse” of preferences; instead, the preference entropy gradually decreases as the critic becomes more informative. We have clarified these dynamics and added a brief discussion of cold-start behavior in `Appendix D.5`.
>
> >   **Q4:** Theoretical vs. practical gap in the prior
>
> **R5:** Thank you for pointing out the discrepancy between the conceptual generative story and the concrete implementation. In the **conceptual generative perspective**, we indeed assume a non-stationary latent dynamics $\omega^\ast_t$ driven by unmodeled situational factors $\xi_t$. In the current implementation, however, we do not explicitly parameterize $p(\omega_t | \omega_{t−1}, \xi_t)$ in the ELBO. Instead, temporal dependencies are encoded **implicitly** via:
>
> -   the **recurrent encoder (GRU)** that conditions $q_\phi(\boldsymbol z_t | s_{t−H+1:t})$ on a history window, and
> -   the **unit Gaussian prior p₀(z)**, which acts as a simple regularizer that discourages abrupt, unsupported jumps in latent preference space.
>
> We agree that explicitly parameterizing $p(\omega_t | \omega_{t−1}, \xi_t)$ in the ELBO is a natural and exciting extension, and we will clarify in the paper that an explicit temporal prior is left for future work.
>
> We chose current design for two reasons: (i) it keeps the optimization problem tractable and stable; and (ii) it allows history-dependent preference dynamics to be represented entirely in the amortized posterior, without introducing an additional transition model to learn. The details can be found in `Appendix C.2`.
>
> We thank you again for the constructive feedback and hope that the additional baselines and analyses will address your concerns. We would be happy to provide further clarification if needed.

---

### Official Review · Reviewer_b9am · 2025-11-01

**Soundness:** 2
**Presentation:** 1
**Contribution:** 3
**Rating:** 2
**Confidence:** 4

**Summary:**

This paper studies the problem of the shifted preference as time and condition/state change. This problem is plural and complex, and the authors proposed a computational agent that comprises the Value Appraisal and Action Selection to model the dynamic decision-making systems.

**Strengths:**

- The paper identifies an interesting problem that may not be fully aware of in the machine learning literature. This problem itself has connections with reinforcement learning and MDPs.
- The proposed framework is quite interesting, and the Bayesian view of this problem is natural to follow. I like how the value shift is integrated in the framework, and I also feel that using the sample example for explaining the framework helps a lot.

**Weaknesses:**

- The paper starts with a lot of terminology but lacks an explanation of these terms. Although I'm quite familiar with the choice model/psychological literature, it is hard to tell if these terms are made up/invented by the authors. I strongly suggest avoiding overuse or misuse of terminology and approaching from an easy-to-understand tone. Besides, proper references for the well-defined terms are needed.
-- I pointed out some points in the Questions section, but there are many more confusing points.

- The paper does not highlight the computational contribution and what makes any other approach hard to address the research question.

I find it quite hard to raise the score to a positive one because of the whole writing and idea delivery, but I would not mind giving a borderline reject if the author could clarify and improve the flow.

**Questions:**

Introduction:
- The statement of Line 28 needs some (psychological) literature to support. The instance given afterward does not fully support the claim "rarely pursuing goals with fixed and immutable priorities".
- In line 36-38, I don't see a clear necessity between the modeling of such dynamic value adaptation and artificial intelligence. Why use AI in particular; why not some other type of modeling? The motivation for using AI is not clear in the writing. You may want to expand a bit on why current literature fails in the computational aspect and why AI can be helpful.
- In line 32-34, I don't think "psychology" and "cognitive science" are two distinct/mutually exclusive terms. They might refer to similar stuff.
- In line 50/51, what does "decision-making models" exactly mean?

---

> ### Author Response · Authors · 2025-11-21
> **Response to Reviewer b9am[1/2]**
>
> We sincerely thank the reviewer for the careful reading, for appreciating the problem we study and the DPI framework, and for the candid feedback on terminology and clarity. We fully agree that the manuscript can be made clearer and more accessible, and that the computational contribution should be highlighted more prominently. We have substantially revised the Introduction and early sections accordingly.
>
> >   **W1:** Use of terminology and psychological references.
>
> **R1:** We apologize that the original draft overused terminology without sufficiently clarifying which notions are standard in the literature and which are specific to our framework.
>
> In the revision, we have made the following changes:
>
> -   **Greatly simplified the terminology in the Introduction.**
>
>     We now describe the phenomenon in plain language (“people juggle multiple goals and adjust their priorities as circumstances change”) instead of introducing a new phrase like “adaptive value preference adjustment” as a central term. We focus on a small set of core notions (e.g., “multiple goals”, “dynamic reweighting of priorities”, “preference weights”) and provide an intuitive gloss when each is first introduced.
>
> -   **Clearly distinguish standard concepts from our own labels.**
>
>     We clarify that terms such as self-regulation, multiple-goal pursuit, and constructed preferences are standard in cognitive and social psychology and cognitive science, and we cite canonical sources. By contrast, we explicitly say that we use “value appraisal” and “action selection” purely as **labels for components of our computational framework**, not as claims about psychological taxonomy. This is now explained in the second paragraph of the Introduction.
>
> -   **Provide proper and more focused references.**
>
>     For each psychological construct we retain, we now include representative, high-impact references. At the same time, we removed less central terminology and moved broader cognitive-science discussion to a more compact, clearly labeled part of the Introduction, so that the main narrative reads more naturally for an ML audience.
>
> Our goal with these changes is that a reader familiar with choice models and MDPs can follow the paper without needing deep expertise in psychology, and can easily see which terms are standard and which are our own architectural names.
>
> >   **W2:** Computational contribution and why other approaches are insufficient.
>
> **R2:** We appreciate the concern that the computational novelty and the role of AI were not sufficiently emphasized in the original version.
>
> To address this, we have:
>
> -   **Made the computational question explicit in the Introduction.**
>
>     Right after summarizing the psychological motivation, we now pose a concrete modeling question:
>
>     “Given only vector-valued rewards and partial observations in a non-stationary environment, how can an agent infer and adapt its current trade-off over objectives online in a way that is both effective and interpretable?”
>
>     This frames the problem clearly as one of **online preference inference** for an artificial agent in a multi-objective, potentially non-stationary environment.
>
> -   **Added a dedicated “why AI / RL” paragraph.**
>
>     We now explicitly explain why we adopt a reinforcement-learning formulation instead of a purely descriptive psychological model. In particular, we note that many psychological and choice-theoretic accounts provide rich descriptions of how humans adjust goals and preferences, but are not directly formulated as algorithms that can be deployed in high-dimensional, partially observable, and non-stationary control problems. In contrast, artificial agents in safety-critical domains must continually act from raw observations under changing constraints, making RL a natural language in which to instantiate dynamic value adaptation.
>
> -   **Highlight what is new computationally about DPI.**
>
>     We now state on the first page that our contribution is to (i) introduce a **new setting** in multi-objective decision-making where preference weights are treated as **latent, dynamically varying states** to be inferred rather than fixed inputs, and (ii) propose **Dynamic Preference Inference (DPI)** as a concrete architecture that:
>
>     1.  maintains a posterior over latent preference weights via an amortized encoder of recent history,
>     2.  conditions an actor–critic on these inferred preferences using an envelope operator, and
>     3.  regularizes preference updates via a prior and alignment terms to keep them stable and interpretable.
>
> We believe these changes make the computational contribution and the reason for using an AI / RL framework much clearer.

---

> ### Author Response · Authors · 2025-11-21
> **Response to Reviewer b9am[2/2]**
>
> >   **Q1:**“Rarely pursuing goals with fixed and immutable priorities.”
>
> **R3:** We agree that the original sentence (“Humans are natural value-driven beings, rarely pursuing goals with fixed and immutable priorities.”) was too strong and not sufficiently supported by the subsequent example.
>
> In the revised Introduction, we **soften and clarify** the claim and ground it more explicitly in established literatures. We now write that human behavior is modeled as goal-directed and value-driven, and that people **typically juggle multiple goals and adjust their priorities as circumstances change**, rather than acting on a single fixed priority ordering. We support this with references on:
>
> -   self-regulation and feedback-based goal control (Carver & Scheier, 2004),
> -   dynamics of multiple-goal pursuit (Louro et al., 2007),
> -   goal adjustment capacities (Wrosch et al., 2003), and
> -   context-dependent, constructed preferences (Slovic, 1995; Lichtenstein & Slovic, 2006).
>
> We also clarify that the queueing example is intended as an **illustrative toy scenario**, not as empirical evidence, and we now rely on the above literatures to justify the broader claim.
>
> >   **Q2:** Why AI / RL for modeling dynamic value adaptation?
>
> **R4:** Thank you for raising this important point. As noted under W2, we have added a dedicated paragraph in the Introduction that explains our modeling choice more clearly. We do not claim that DPI is a veridical model of human cognition; rather, we use cognitive theories as inspiration for a computational architecture tailored to RL settings.
>
> In brief, our interest is in **artificial agents that must act in complex, multi-objective environments** (e.g., safety vs. efficiency, energy vs. morality) where constraints and trade-offs can change over time. In such settings, the agent must not only represent dynamic preferences but also use them to control behavior online from high-dimensional, partially observable inputs.
>
> We therefore treat dynamic value adaptation as a **computational control problem** and formulate it in the language of multi-objective reinforcement learning. We explicitly contrast this with many psychological models, which are primarily descriptive and not directly designed to operate as scalable control algorithms in such environments. The revised text makes this motivation explicit and states that DPI should be viewed as a bridge between cognitive theories of goal regulation and practical RL-based controllers.
>
> >   **Q3:** “Psychology” vs. “cognitive science” as two distinct terms.
>
> **R5:** We agree that our original wording could be read as treating “psychology” and “cognitive science” as distinct or mutually exclusive fields, which was not our intention. In the revision, we have simplified the phrasing and now refer to “work in cognitive and social psychology and cognitive science” without implying a sharp boundary between these communities. Our goal is simply to acknowledge that related ideas come from both psychological and broader cognitive-science traditions.
>
> >   **Q4:** Meaning of “decision-making models.”
>
> **R6:** Thank you for pointing out that this phrase was too vague. In the revised Introduction, we replace “decision-making models” with a more precise expression:
>
> **computational models of sequential decision-making (including classical control systems, Markov decision processes, and modern reinforcement learning agents)**
>
> and we explicitly distinguish these from psychological and choice-theoretic models. We also clarify that our critique is aimed at models that assume a **fixed, externally specified utility or reward function**, in contrast to our setting where preference weights are latent and dynamically inferred.
>
> We hope that these revisions address your concerns about terminology, motivation, and clarity of the computational contribution. We are grateful for your feedback, which helped us significantly improve the flow and accessibility of the paper, and we would be happy to further clarify any remaining points.

---

### Official Review · Reviewer_tNvs · 2025-11-02

**Soundness:** 2
**Presentation:** 3
**Contribution:** 2
**Rating:** 4
**Confidence:** 2

**Summary:**

The paper considers dynamically changing preferences. It proposes an algorithm that leverages a signal providing information about these signals to modify its actions accordingly. Empirical results are presented on two settings (waiting in a queue, a maze problem) comparing the method to baselines.

**Strengths:**

The paper considers an interesting under-explored area. There is strong empirical evidence that humans switch their stated goals over time. There is value in understanding this phenomenon and in designing algorithms that support humans despite these changes.

**Weaknesses:**

The motivation for the paper is unclear (see questions #1, #2, #3 below).  It is also unclear how to apply it to a practical problem (questions #1, #2, #4).  The experiments should have included stronger RL-based baselines that maximize the performance metrics reported (question #5).

**Questions:**

#1. In the "waiting in line" example used to motivate the paper and used in the numerical experiments, the phenomenon described can be modeled without the need to change goals.
  The goal would be to maximize the expected value of the utility function 1{waiting_time < T} + c*cut_in_line.
  The optimal strategy is to wait until T and, if we haven't gotten to the front of the line, cut.
  The thing that is changing is not the goal but the optimal action (cut or not cut) given the current state (how long we've been waiting in line).

Let me explain my example in more detail so that the authors can respond.  My example is a discrete time problem where there is a random variable W that is the time we would wait if we chose not to cut in line. This is drawn from a known probability distribution.  In each time period t, if we haven't gotten to the front of the line yet (i.e., t < W), then we would decide whether to cut in line or not based on t.  Any decision rule (a way to decide the action, i.e., whether to cut, based on the current state, i.e., t) results in a binary random variable cut_in_line, which is 1 if we eventually choose to cut in line and 0 if not. It is a function of W.  A decision rule also results in a waiting time, which is the minimum of W and when we choose to cut in line, if ever.  Using a dynamic programming argument where the state is the current time t, you can show that if c is large enough, an optimal strategy will be to wait until time T-1 and, if we haven't gotten to the front of the line, cut.

These same arguments extend to utility functions of the form f{waiting_time} + c*cut_in_line for a general function f. For most functions f and distributions over W, the optimal strategy will be to wait until some finite time that is strictly larger than 0 and then cut.

This is an optimal stopping problem.

#2. The maze problem (the second numerical example) also seems to be better modeled by a fixed goal. What the paper positions as changes in goal really just seem like changes in the state in a Markov decision process (MDP) or partially observable MDP. For example, when trying to get out of a maze, a reasonable fixed goal is to minimize the expected time to get out of the maze. Over time, our knowledge about the maze changes. The maze itself may also change. As a result, the optimal action may change.

#3. I am having trouble understanding the motivation for the paper.  I agree that there is a large body of evidence showing that humans "switch goals" over time. But this doesn't imply that algorithms should do this.

Is the goal to develop a decision-making algorithm that is somehow better able to provide value to humans, in light of this empirically-observed goal switching?  Or is it to improve our understanding of human decision-making?

My sense is that the paper's goal is the not the second --- if it were, then we would presumably want to include human data in the experiments and make comparisons to other hypotheses.
Presumably then it is the first.  But if this is the case, then it isn't clear that we want our algorithms to "swich goals".

First, for settings like my comments #1 and #2 where human "goal switching" can be explained as the use of decision-making shortcuts to perform well against a larger goal that is static, wouldn't human utility be better maximiized by doing a good job of optimizing against the larger goal with a gool RL strategy?  More broadly, if "goal switching" can be explained by humans' bounded rationality because of biological limitations, then wouldn't a human prefer that an algorithm making decisions on their behalf be more rational than they are?

I am not philosophically opposed to developing an algorithm for settings where humans switch goals but to be publishable at ICLR there would need to be a stronger set of examples where it is demonstrated that the new algorithm provides value to a human. To demanstrate that it provides more value to a human, it is important to somehow bring human data in to the paper, e.g., by having a human assess the quality of the outcomes provided by the proposed algorithm and baselines in a real-world setting.

#4. Practically, how should a practitioner interested in using this method estimate p(s_t | omega_t) and p(omega_t | omega_{t-1}, xi_t)?

#5. In the experiments, methods are evaluated relative to static success measures --- mean episodic return, success rate, and post-shift performance at K. If I want to maximize some combination of these (a weighted combination, or maximizing one subject to constraints on the others), then I should use a reinforcement learning algorithm designed for this task. The Queue environment seems particularly simple and I would expect that an RL algorithm could accomplish this. At a minimum, the paper should include as a baseline an RL algorithm that optimizes each of the single metrics reported.

#6. Please help me understand some of the notation in section 3.1.
- In equation 1, which is s_t not on the right-hand side of the equation?
- In equation 1, presumably omega is a function of z_t even though the notation from equation 2 indicating this is not used here.  This notation from equation 2 is useful --- it should be introduced and defined in the text.
- In equation 2, why does s_{t-H+1:t} not appear on the right-hand side?

---

> ### Author Response · Authors · 2025-11-21
> **Response to Reviewer tNvs[1/3]**
>
> We sincerely thank the reviewer for the careful reading and for articulating the concerns so clearly. We found the comments very helpful in sharpening both the motivation and the technical presentation of the paper. Below we address each point in turn and have incorporated the clarifications into the revised manuscript.
>
> >   **W1&Q1&Q2:** Unclear motivation. Why dynamic preferences? Why not a fixed goal with optimal stopping or MDP planning?
>
> **R1:** We fully agree that **if the agent has access to the true environment model and the correct reward scalarization**, then both the Queue and the Maze can be solved using standard dynamic programming or RL with a fixed utility function. From an omniscient designer’s perspective, one could encode the different regimes into a single global utility function, and, as the you notes, if the distribution of the waiting time $W$ and the scalar utility are known, the optimal “wait-until-$T$-then-cut” policy follows from an optimal stopping argument.
>
> Our setting differs in one crucial aspect:
>
> **the event-driven changes that determine the correct trade-off between objectives are latent and not externally provided to the agent.**
>
> Concretely:
>
> -   the agent **does not** have access to the true scalar utility or the distribution of $W$;
> -   the environment provides **vector-valued feedback** (e.g., progress, fairness, energy), but the relative importance of each component is **latent and may drift** with context;
> -   the agent must act **under bounded rationality** with limited knowledge and working memory.
>
> Empirically, in our instantiations the Pareto-optimal preference vectors that maximize performance before and after an event have **very limited overlap**:
> $$
> \arg\max_{\boldsymbol \omega}\langle \boldsymbol\omega, \vec{G}_{\text{pre}} \rangle  \neq  \arg\max _{\boldsymbol \omega} \langle  \boldsymbol \omega , \vec{G} _{\text{post}} \rangle,
> $$
> so any single fixed $\boldsymbol{\omega}$ incurs sharp trade-offs across event configurations (see `Appendix D.1` for a visualization of pre- and post-event Pareto fronts). In other words, while a global utility may exist in principle, at the level of the observed vector rewards any static scalarization is **empirically brittle** under latent regime shifts.
>
> DPI is designed specifically for this scenario: **the agent must infer a latent preference state** $\boldsymbol{\omega}^{*}_t$ **from partial evidence and adjust its behavior accordingly**, rather than committing to a static trade-off.
>
> We will clarify this distinction explicitly in Sec. 3.
>
> >   **Q3**: Motivation problem: Is “goal switching” intended to model human bounded rationality?
>
> **R2:** We apologize for the confusion. The human examples in the introduction are intended purely as intuition, not as a psychological model we claim to fit.
>
> Algorithmically:
>
> -   $\omega_t^{*}$ is treated as a **latent variable** representing the current (unobserved) reward trade-off induced by an event.
> -   The encoder learns an **approximate posterior** $q(\omega_t \mid s_{t-H+1:t})$.
> -   The policy selects actions that **maximize expected utility under the inferred preference**.
>
> Our aim is not to mimic human bounded rationality, but to design agents that can achieve high utility when the true scalarization is unknown and effectively non-stationary from the agent’s perspective.
>
> We will simplify the psychological terminology in the introduction to avoid suggesting a stronger cognitive claim.

---

> ### Author Response · Authors · 2025-11-21
> **Response to Reviewer tNvs[2/3]**
>
> >   **W2&Q4**: Practical interpretation of $p(s_t \mid \omega_t)$ and $p(\omega_t \mid \omega_{t-1},\xi_t)$
>
> **R3:** Thank you for highlighting this. Our intention with the “cognitive & statistical generative assumption” was to provide a conceptual Bayesian perspective on preference dynamics, rather than to require practitioners to explicitly parametrize and estimate these two conditional distributions.
>
> In the practical implementation used in our experiments:
>
> -   We introduce an unconstrained latent variable $z_t$ with a simple prior $p_0(z_t) = \mathcal N(0,I)$ and map it to preference weights via $\omega_t = f_\theta(z_t)$ (softmax over logits).
>
> -   The environment itself determines the dynamics over states and rewards; we do **not** fit a separate environment model. Instead, we define an **implicit Boltzmann-rational likelihood** of the evidence $e_t := s_{t-H+1:t}$ via the scalarized return:
>     $$
>     p(e_t \mid \boldsymbol{z}_t) \propto \exp\big(\beta U_t(\boldsymbol{\omega}_t; e_t)\big), \quad U_t(\boldsymbol{\omega}_t; e_t) = \langle \boldsymbol{\omega}_t, \vec G_t(e_t)\rangle,
>     $$
>     where $\vec G_t(e_t)$ is the vector return estimated by the critic from the observed history. This plays the role of $p(s_t \mid \omega_t)$ in the generative story.
>
> -   The **temporal dependency** over preferences, which in the conceptual model is written as $p(\omega_t \mid \omega_{t-1}, \xi_t)$, is implemented in practice by an *recurrent encoder* $q_\phi(z_t \mid e_t)$ that takes a short history window $e_t = s_{t-H+1:t}$ as input. That is, instead of specifying a parametric transition prior over $\omega_t$, we let the encoder learn how beliefs should be updated from recent observations.
>
> **Thus, a practitioner using DPI does not need to explicitly estimate $p(s_t \mid \omega_t)$ or $p(\omega_t \mid \omega_{t-1}, \xi_t)$.** In practice, they only need to:
>
> 1.  Choose a recurrent encoder architecture for $q_\phi(z_t \mid s_{t-H+1:t})$;
> 2.  Use the Boltzmann-rational utility model above to define the ELBO;
> 3.  Train the encoder and preference-conditioned actor–critic jointly using the combined RL + ELBO objective.
>
> We will clarify this point in the main text.
>
> >   **Q5**: RL baselines optimizing static metrics.
>
> Thank you for this concrete suggestion. We realize that part of the confusion comes from how MER / SR / PS@K are meant to be used in our setting. In our formulation, the environment exposes vector-valued feedback, and the agent learns a latent, dynamic scalarization $\omega_t$. MER, SR, and PS@K are then used by an external evaluator to probe different aspects of the learned behavior (overall utility, value-consistent episodes, post-shift recovery), but they are not the rewards given to the agent during training.
>
> Training a separate RL policy that “optimizes MER” or “optimizes SR” therefore changes the task: it requires us to redesign a scalar reward function tailored to each metric. For example, an SR-optimized agent receives a sparse success reward (1 on successful episodes, 0 otherwise), effectively ignoring the underlying multi-objective structure.
>
> Following your suggestion, we implemented two such “single-metric” RL baselines:
>
> -   MER-PPO: PPO with a metric-specific scalar reward $r_t^{\text{MER}} = \langle \omega^{\text{eval}}, \vec r_t \rangle,$ where the evaluation weight $\omega^{\text{eval}}$ is given to the agent (oracle static scalarization).
>
> -   SR-PPO: PPO with a sparse success reward (1 on success, 0 otherwise).
>
> On the Queue environment, we obtain:
>
> | Method  | Queue           |                 | Maze            |                |
> | ------- | --------------- | --------------- | --------------- | -------------- |
> |         | **MER**         | **SR (%)**      | **MER**         | **SR(%)**      |
> | MER-PPO | $15.01\pm0.46$  | $0.98\pm 0.05$  | $85.55\pm0.12$  | $0.05\pm0.01$  |
> | SR-PPO  | $-5.64\pm5.22$  | $46.92\pm 0.00$ | $-15.28\pm0.96$ | $61.13\pm0.05$ |
> | DPI-PPO | $10.34\pm 0.02$ | $39.95\pm 2.75$ | $30.16\pm1.22$  | $59.04\pm0.01$ |
>
> where DPI-PPO denotes our proposed DPI combined with PPO under the same vector-valued feedback, numbers of SR are in $\%$. As reported, each of these agents improves its target metric but at the cost of the others. In detail, SR-PPO increases SR but substantially degrades MER while MER-PPO over-optimizes scalar return and performs poorly on SR (i.e., it rarely succeeds on both tasks).
>
> We have included these results in `Appendix~D.2` of the revised paper.

---

> ### Author Response · Authors · 2025-11-21
> **Response to Reviewer tNvs[3/3]**
>
> >   **Q6**: Unclear notations in Section 3.1
>
> **R5:** Thank you for pointing out the confusing notation. You are absolutely right that our current presentation makes the dependence on the evidence $s_{t-H+1:t}$ and on $z_t$ unnecessarily implicit.
>
> **(a) Which is $s_t$ not appear on the RHS of Eq. (1)?**
>
> In the original version, the dependence on the state history was hidden inside the utility term $U_t$ through the vector return $\vec G_t$. This makes it look as if the likelihood did not depend on the evidence at all. In the revised manuscript, we explicitly introduce $e_t := s_{t-H+1:t}$ as the evidence and rewrite Eq. (1) as
> $$
> p(e_t \mid z_t) \propto \exp\big(\beta\, U_t(\omega_t; e_t)\big), \quad \beta > 0,
> $$
> where $U_t(\omega_t; e_t) = \langle \omega_t, \vec G_t(e_t)\rangle.$ This makes the dependence on the observed history explicit.
>
> **(b) $\omega$ as a function of $z_t$.**
>
> We now explicitly define $\omega_t = f_\theta(z_t)$ (e.g., a linear map followed by softmax) and use this notation consistently in all equations, so that both Eq. (1) and Eq. (2) refer to $U_t(\omega_t)$ with $\omega_t = f_\theta(z_t)$.
>
> **(c) Why does $s_{t-H+1:t}$ not appear on the RHS of Eq. (2)?**
>
> In the revised version, we make the Bayes relationship explicit: $p^{\ast} (\boldsymbol{z}_t \mid e_t) \propto p_0(\boldsymbol{z}_t) \cdot \exp\big(\beta U_t(\boldsymbol\omega_t; e_t)\big).$
>
> where the evidence $e_t$ influences the posterior through $U_t$ and $\vec G_t(e_t)$. The conditional $q_\phi(\boldsymbol{z}_t \mid e_t)$ and the ELBO are written with $e_t$ as an explicit conditioning variable.
>
> We hope this addresses the confusion about where the information from $s_{t-H+1:t}$ enters the posterior. We will also update `Sec 3.1` and `Appendix A`, so that the probabilistic structure is fully transparent.
>
> Thanks for the valuable feedback and hope that the above clarifications resolve your concerns. We would be happy to further discuss any remaining issues.

---

### Author Response · Authors · 2025-12-01
**Author Summary to the Area Chair**

We sincerely thank the AC and the reviewers for their time and thoughtful feedback. Below we (i) explain how the revision addresses the main concerns and (ii) briefly summarize the strengths highlighted in the reviews.

---

We have substantially revised the manuscript along three directions:

**1. Problem setting, motivation, and terminology (Reviewers #tNvs, #b9am)**

We clarified the distinction between an omniscient modeller and the agent’s perspective with **latent, context-dependent preference weights inferred online**, refined the motivation for dynamic preferences in the Queue and Maze settings (including a pre-/post-event Pareto-front illustration), and simplified the terminology to foreground the **computational framework** rather than psychological labels.

**2. Baselines, experimental scope, and stability (Reviewers #tNvs, #6Kvv, #9DQE)**

We strengthened the empirical section by clarifying the role of **fixed-scalarization RL baselines**, adding a **Dense Oracle baseline** with privileged event information, introducing a **continuous-control multi-objective MuJoCo-style environment**, and reporting MER/SR with confidence intervals and **training-dynamics / hyperparameter-sensitivity analyses**.

**3. Generative assumptions, inference, and notation (Reviewers #tNvs, #6Kvv)**

We made explicit that we use an **amortized recurrent inference network** instead of learning full transition models, explained how temporal dependence among preferences is captured in practice, and revised Eqs. (1)–(2) and surrounding text to clarify conditioning and notation.

---

**Recognized strengths of the work (as noted by the reviewers)**

- Novel perspective on dynamic preferences and problem formulation. (Reviewers #tNvs, #b9am, #6Kvv, #9DQE)
- Clear organization and presentation. (Reviewer #9DQE)
- Principled probabilistic framework with interpretability. (Reviewer #6Kvv)
- Informative experiments and ablations. (Reviewers #6Kvv, #9DQE)

---

**Factors we hope the AC will consider**

In light of the reviews and the revisions, we kindly ask the AC to consider:

1. **Novelty and conceptual contribution.**
    The paper proposes a dynamic preference inference framework where preference weights are latent, context-sensitive states inferred online. Reviewers have found this perspective novel and interesting, with clear links to both cognitive theories and multi-objective RL.
2. **Sound and interpretable computational framework.**
    The architecture (variational appraisal + preference-conditioned actor–critic + envelope selection) is technically sound, implementable, and interpretable, enabling inspection of how inferred preferences evolve with events.
3. **Strengthened empirical support.**
    The revised version includes stronger baselines (including a Dense Oracle), a more realistic continuous-control environment, training-stability and sensitivity analyses, and clearer exposition and notation that directly address the main reviewer concerns.

We hope this summary helps the AC in evaluating the revised manuscript.

---

### Meta-Review · Area_Chair_zM6P · 2026-01-06

**Summary:**

This paper addresses the challenge of multi-objective reinforcement learning (MORL) in non-stationary environments where the relative importance of objectives (preference weights) is not fixed or observed, but is a latent variable that drifts based on context. The authors propose "Dynamic Preference Inference" (DPI), a cognitively inspired framework that uses a variational inference module to maintain a probabilistic belief over these latent preferences based on recent history. This inferred belief conditions an actor-critic policy, utilizing an on-policy envelope operator to select optimal preference candidates. The method is evaluated in queueing and grid-world environments, and importantly, a continuous control MuJoCo environment added during the rebuttal phase.

Identified Upon reviewing the submission and the initial feedback, the central concern was whether the proposed machinery was justified given the simplicity of the evaluation. Specifically, the initial reliance on toy domains (Queue and Maze) raised valid doubts about the method's scalability and whether simpler "heuristic" or "fixed" baselines were strawmen. Furthermore, there was a fundamental question regarding motivation: reviewers questioned if the "dynamic preference" framing was distinct enough from standard POMDPs or if a generic recurrent policy could solve the task implicitly without the specific inference architecture. The lack of strong "Oracle" baselines or agents optimizing specific static metrics further weakened the claim that dynamic inference was necessary.

I recommend (weak) acceptance as a poster. The authors provided a comprehensive and highly effective rebuttal that directly addressed the objective weaknesses of the initial submission. Crucially, they implemented a new continuous control environment (Multi-objective HalfCheetah) and added the requested "Dense Oracle" and single-metric optimization baselines. These additions demonstrated that the method provides value beyond toy settings and outperforms strong static scalarizations. While some theoretical questions remain regarding whether generic recurrent policies could eventually learn similar behaviors, the proposed variational approach offers a principled and interpretable alternative that is now empirically well-supported. The paper makes a solid contribution to the MORL literature in non-stationary settings.

**Reviewer Concerns:**

Original reviewer concerns focused on motivation (e.g. tNvs: "is this just a POMDP?"), presentation (e.g. b9am: "confusing terminology"), baselines (see tNvs and 6Kvv), and experimental scope (see 6Kvv and 9DQE: "toy domains only"). The authors provided a comprehensive rebuttal that addressed the majority of substantive concerns:

* A major concern was experimental scope. In response to Reviewers 6Kvv and 9DQE, the authors implemented a new Continuous Control environment (Multi-objective HalfCheetah with dynamic events). This significantly strengthened the empirical validation.
* Having the right baselines was also a major concern. The authors added:
    * Dense Oracle: A PPO agent with privileged access to event dynamics (the "upper bound").
    * Static Metric Optimisers (MER-PPO/SR-PPO): Demonstrating that optimising for a single static scalarization fails to capture the trade-offs required across regimes.
* Regarding motivation and terminology, the authors clarified the distinction between an omniscient designer vs. an agent with limited knowledge (pre/post-event Pareto analysis) and rewrote the introduction to remove confusing psychological claims.

However, some concerns remained outstanding:

* First was the comparison to generic recurrent policies. While the authors compared against "Fixed" and "Heuristic" baselines, the paper does not definitively rule out whether a generic Recurrent PPO (without the specific variational inference module) could implicitly learn to handle the latent shifts. This leaves the "necessity" of the specific DPI architecture slightly open.
* Also, the inference module relies on vector-valued returns estimated by the critic. In sparse reward or hard-exploration settings, noisy critic estimates could destabilise the preference inference, a limitation that remains relevant for future work.

**Reviewer Scores:**

**tNvs (Original: 4): Estimated New Score: 6**: The reviewer’s specific requests for baselines were met. While they may still have doubts about the "dynamic preference" framing vs. standard POMDPs, the technical objections preventing acceptance have been removed. A move to "Marginally Above" is the most rational outcome.

**Reviewer b9am (Original: 2): Estimated New Score: 4**: This reviewer started at a "Reject" (2). While the authors fixed the terminology and presentation, jumping all the way to an "Accept" (6 or 8) is rare for a reviewer who started so negatively. A move to 4 acknowledges the paper is now "readable/valid" but reflects likely residual lack of enthusiasm.

**Reviewer 6Kvv (Original: 4): Estimated New Score: 6**: This reviewer was the most constructive, asking for specific, heavy-lift experiments (Continuous Control, Dense Oracle). The authors delivered 100% on these requests.

**Reviewer 9DQE (Original: 6): Estimated New Score: 8**: Already positive; the new experiments solidify the work.

---

### Decision · Program_Chairs · 2026-01-26

Accept (Poster)